# Antibody targeting of E3 ubiquitin ligases for receptor degradation

Hadir Marei[1,13], Wen-Ting K. Tsai[2,13], Yee-Seir Kee[2,13], Karen Ruiz[1], Jieyan He[3], Chris Cox[4], Tao Sun[5], Sai Penikalapati[6], Pankaj Dwivedi[6], Meena Choi[6], David Kan[7], Pablo Saenz-Lopez[7], Kristel Dorighi[5], Pamela Zhang[2], Yvonne T. Kschonsak[1], Noelyn Kljavin[8], Dhara Amin[1], Ingrid Kim[2], Andrew G. Mancini[8], Thao Nguyen[8], Chunling Wang[1], Eric Janezic[3], Alexander Doan[3], Elaine Mai[3], Hongkang Xi[9], Chen Gu[10], Melanie Heinlein[8], Brian Biehs[8], Jia Wu[9], Isabelle Lehoux[11], Seth Harris[12], Laetitia Comps-Agrar[3], Dhaya Seshasayee[9], Frederic J. de Sauvage[8], Matthew Grimmer[1], Jing Li[3], Nicholas J. Agard[2✉] & Felipe de Sousa e Melo[1✉]

Most current therapies that target plasma membrane receptors function by antagonizing ligand binding or enzymatic activities. However, typical mammalian proteins comprise multiple domains that execute discrete but coordinated activities. Thus, inhibition of one domain often incompletely suppresses the function of a protein. Indeed, targeted protein degradation technologies, including proteolysis-targeting chimeras[1] (PROTACs), have highlighted clinically important advantages of target degradation over inhibition[2]. However, the generation of heterobifunctional compounds binding to two targets with high affinity is complex, particularly when oral bioavailability is required[3]. Here we describe the development of proteolysis-targeting antibodies (PROTABs) that tether cell-surface E3 ubiquitin ligases to transmembrane proteins, resulting in target degradation both in vitro and in vivo. Focusing on zinc- and ring finger 3 (ZNRF3), a Wnt-responsive ligase, we show that this approach can enable colorectal cancer-specific degradation. Notably, by examining a matrix of additional cell-surface E3 ubiquitin ligases and transmembrane receptors, we demonstrate that this technology is amendable for 'on-demand' degradation. Furthermore, we offer insights on the ground rules governing target degradation by engineering optimized antibody formats. In summary, this work describes a strategy for the rapid development of potent, bioavailable and tissue-selective degraders of cell-surface proteins.

Over the past two decades, important strides have been made in developing small molecule-based protein degraders[1,4,5]. Typically, these involve heterobifunctional molecules such as PROTACs[1] and molecular glues[6] that form a ternary complex with an E3 ubiquitin ligase and a target of interest, resulting in target ubiquitination and degradation. Although effective biochemically, the therapeutic potential of PROTACs has been hampered by the poor permeability, pharmacokinetics and pharmacodynamics properties commonly seen with high molecular mass (over 1,000 Da) small molecules[3,7]. More recently, large molecule-based degrader technologies, including Trim-away[8,9] and lysosome targeting chimeras[10] (LYTACs) have highlighted the potential of leveraging large molecules for targeted protein degradation. Building on these discoveries, we set out to explore intrinsic cellular degradative machinery that could be repurposed for an antibody-based targeted protein degradation platform and identified membrane-bound E3 ubiquitin ligases with exposed extracellular domains (ECDs) as attractive candidates.

### Identification of Wnt-responsive ligases

Among the cell-surface E3 ubiquitin ligases, we focused on ring finger protein 43 (RNF43) and ZNRF3, two known negative regulators of Wnt signalling that promote the turnover of Wnt receptors Frizzled/low-density lipoprotein receptor-related proteins (FZD/LRPs) via ubiquitination-mediated degradation[11,12]. These proteins ensure proper regulation of Wnt activity in normal adult stem cells. Given that RNF43 and ZNRF3 are downstream of Wnt signalling, we anticipated that their expression would selectively increase in Wnt-hyperactive disease states. To model aberrant Wnt signalling, we generated mouse

[1]Discovery Oncology, Genentech, South San Francisco, CA, USA. [2]Antibody Engineering, Genentech, South San Francisco, CA, USA. [3]Biochemical and Cellular Pharmacology, Genentech, South San Francisco, CA, USA. [4]Discovery Immunology, Genentech Inc, South San Francisco, CA, USA. [5]Molecular Biology, Genentech, South San Francisco, CA, USA. [6]Microchemistry, Proteomics and Lipidomics, Genentech, South San Francisco, CA, USA. [7]Translational Oncology, Genentech, South San Francisco, CA, USA. [8]Molecular Oncology, Genentech, South San Francisco, CA, USA. [9]Antibody discovery, Genentech, South San Francisco, CA, USA. [10]Protein Chemistry, Genentech, South San Francisco, CA, USA. [11]Biomolecular Resources, Genentech, South San Francisco, CA, USA. [12]Structural Biology, Genentech, South San Francisco, CA, USA. [13]These authors contributed equally: Hadir Marei, Wen-Ting K. Tsai, Yee-Seir Kee. ✉e-mail: agardn@gene.com; desousaf@gene.com

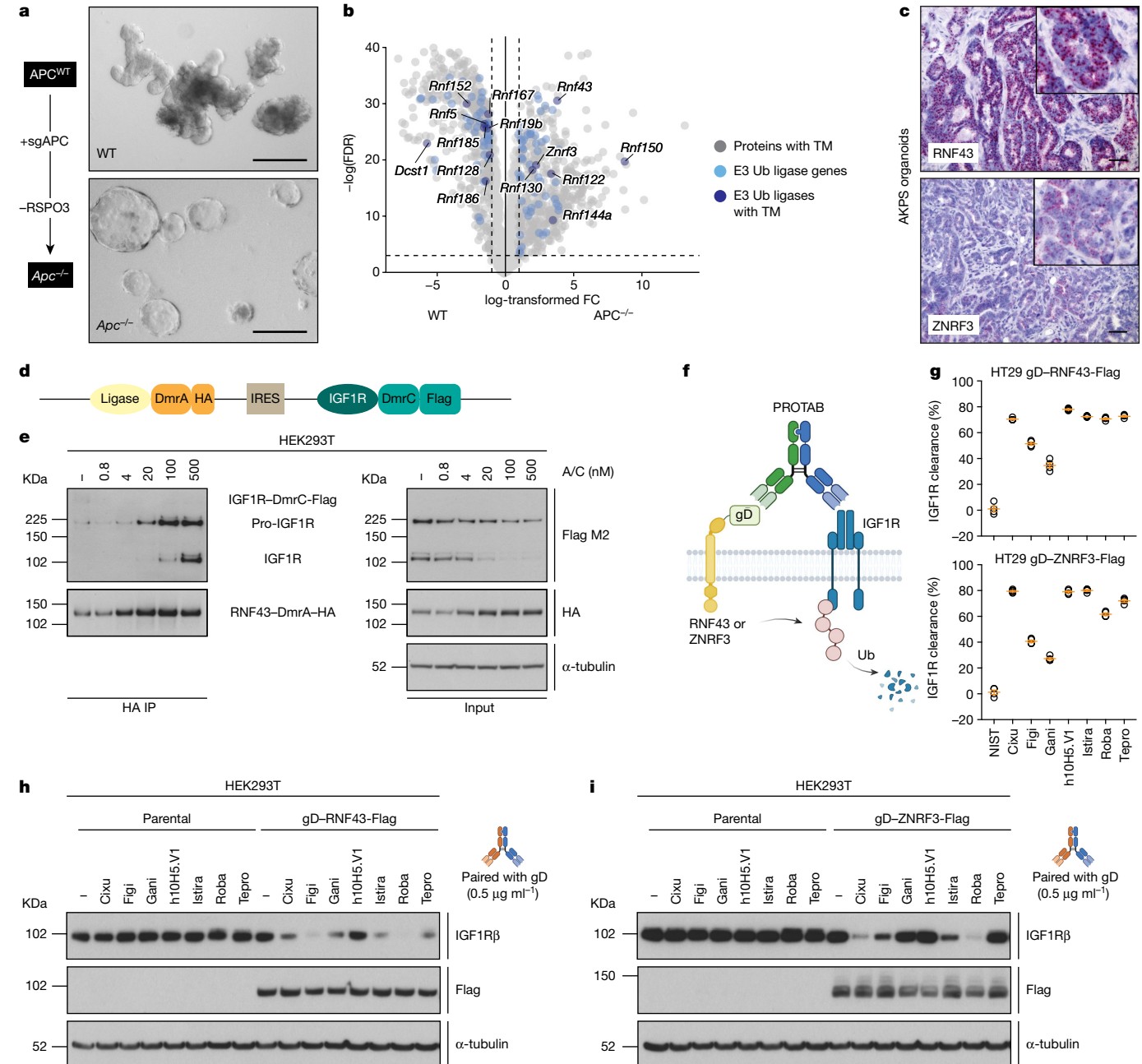

**Fig. 1 | The Wnt-responsive E3 ubiquitin ligases RNF43 and ZNRF3 can degrade IGF1R. a**, Left, CRISPR–Cas9 strategy used to generate APC mutant organoids. Right, phase-contrast images of wild-type (WT) and APC truncation mutant (*Apc*[−/−]) colon organoids. Data are representative of two independent experiments. Scale bars, 250 μm. **b**, Gene expression analysis of wild-type and *Apc*[−/−] colon organoids. Genes encoding proteins with predicted transmembrane domains (TM) (grey), E3 ubiquitin (Ub) ligases (light blue) or both (dark blue) are shown. Data are the mean expression from three independent experiments. FC, fold change; FDR, false discovery rate. **c**, In situ hybridization in AKPS colon organoids probing for *Rnf43* and *Znrf3*. Data are representative of two independent experiments. Scale bars, 50 μm. **d**, Schematic representation of the iDimerize construct. HA, haemagglutinin tag. **e**, Levels of total and immunoprecipitated IGF1R and RNF43 following HA tag immunoprecipitation (IP) following treatment of HEK293T cells harbouring the RNF43–IGF1R iDimerize construct

with the A/C heterodimerizer. **f**, Schematic representation of gD-based PROTAB-mediated IGF1R degradation. **g**, Levels of cell-surface IGF1R assessed by flow cytometry following treatment of HT29 cells expressing gD–RNF43–Flag or gD–ZNRF3–Flag with 10 μg ml[−1] of the indicated gD-based PROTABs for 48 h. Graphs depict IGF1R clearance from four independent experiments. Data are mean ± s.e.m., with values from individual biological repeats overlaid. **h,i**, Levels of IGF1Rβ following treatment of parental HT29 cells or cells expressing gD–RNF43–Flag (**h**) or gD–ZNRF3–Flag (**i**) with 0.5 μg ml[−1] of the indicated PROTABs for 24 h. In **e,h,i**, α-tubulin was used as a loading control. Data are representative of three independent experiments. IGF1R-binding antibodies: Cixu, cixutumumab; Figi, figitumumab; Gani, ganitumab; h10H5.V1; Istira, istiratumab; Roba, robatumumab; Tepro, teprotumumab. For gel source data, see Supplementary Fig. 1.

intestinal organoids containing a frameshift truncation in the adenomatous polyposis coli (*Apc*) gene (Fig. 1a), which leads to Wnt pathway hyperactivation and the initiation of colon cancer[13]. Gene expression analysis of these engineered organoids (*Apc*[−/−]) revealed a significant

increase in the expression of Wnt-related signatures compared with wild-type organoids (Extended Data Fig. 1a). Of note, a number of E3 ubiquitin ligases also exhibited increased expression in *Apc*[−/−] organoids (Fig. 1b). Focusing on E3 ubiquitin ligases with predicted

transmembrane domains, only RNF43 and ZNRF3 displayed a similar pattern of increased expression in human colon adenomas compared with normal tissue (Extended Data Fig. 1b,c). In the normal gut, RNF43 expression is restricted to the bottom of the intestinal crypt where stem cells reside[12]. To compare this expression pattern with that of colorectal cancer (CRC), we further engineered the $Apc^{-/-}$ organoids to add relevant CRC mutations and sequentially introduced $Kras^{G12D}$ (AK), $Trp53$ (AKP) and $Smad4$ (AKPS) gene alterations. In situ hybridization of carcinomas resulting from orthotopic implantation[14,15] of AKPS organoids revealed high-intensity and homogenous staining for both ligases within the tumour (Fig. 1c). Thus, hyperactivation of Wnt signalling in CRC leads to an increase in expression of RNF43 and ZNRF3 extending beyond the restricted expression seen in the corresponding normal tissue.

## RNF43 and ZNRF3 as IGF1R degraders

The near-universal loss of APC in CRC renders tumours Wnt ligand-independent, bypassing the regulatory activities of RNF43 and ZNRF3. We postulated that these ligases could be repurposed to selectively degrade other substrates in Wnt-hyperactivated tumours. As a proof of concept, we focused on the insulin growth factor 1 receptor (IGF1R), a receptor tyrosine kinase that mediates growth factor signalling in various tissues and cancers and has previously been inhibited with therapeutic antibodies[16]. We explored the effect of ligase–IGF1R colocalization by fusing dimerization domains (DmrA and DmrC) to the respective proteins and adding a chemical inducer of dimerization (A/C heterodimerizer) (Fig. 1d and Extended Data Fig. 1d). Treatment with the A/C heterodimerizer induced dose-dependent binding of RNF43–DmrA or ZNRF3–DmrA to IGF1R–DmrC. Notably, heterodimerization was accompanied by a concentration-dependent decrease in protein levels of IGF1R–DmrC (Fig. 1e and Extended Data Fig. 1e).

To translate this finding towards a therapeutic format, we generated bispecific antibodies pairing an antibody arm targeting an N-terminal glycprotein D (gD) epitope to one of seven previously established IGF1R antibodies[16–19], spanning a range of affinities and epitopes (Fig. 1f and Extended Data Fig. 1f,g). Addition of PROTABs to cells expressing gD fused to RNF43 or ZNRF3 (gD–RNF43–Flag or gD–ZNRF3–Flag) induced cell-surface clearance and degradation of IGF1R, whereas cells lacking ectopic ligase expression were not affected (Fig. 1g–i). Collectively, these results show that ligase–target dimerization can mediate target degradation. Of note, the affinity for IGF1R targeting antibody did not fully predict cell-surface clearance, suggesting that the epitope—and therefore the geometry of the ternary complex—may influence target degradation.

To extend this approach to endogenous ligases, we conducted antibody discovery campaigns against the ECDs of RNF43 and ZNRF3. Immunization campaigns led to 40 anti-ZNRF3 and 65 anti-RNF43 antibodies with picomolar to mid-nanomolar affinities and unique (non-sibling) sequences in heavy chain complementarity-determining region 3 (Extended Data Fig. 2a). We paired a subset of these antibodies with cixutumumab (Cixu) (Fig. 2a), selected for its substantial but incomplete activity as a Cixu*gD PROTAB (that is, a PROTAB with Cixu- and gD-recognizing arms). This created a window to further enhance degradation with a ligase-paired bispecific molecule. As a control, we paired Cixu with an irrelevant (NIST) antibody[20]. Cell-surface clearance efficiency of IGF1R in HT29 cells overexpressing the ligases correlated with the affinity of the ligase binding arm, reaching a plateau around 1 nM (Fig. 2b,c). The saturating affinity also approximates that for the binding of the Cixu arm, suggesting that similar to PROTACs, the durability of the ternary complex influences the activity. The activity of endogenous ligases was also assessed in HT29 cells genetically engineered to harbour an N-terminal luciferase tag[21] (HiBiT) on IGF1R (Extended Data Fig. 2b). Using chemiluminescence as a proxy for IGF1R cell-surface levels, numerous IGF1R PROTABs led to substantial

IGF1R cell-surface clearance (Fig. 2d and Extended Data Fig. 2c). Kinetic analysis showed progressive, dose-dependent, and reversible IGF1R degradation, with clearance saturation observed within 24–48 h after treatment (Extended Data Fig. 2d,e). The PROTABs drove efficient IGF1R degradation across a comprehensive panel of CRC lines with various endogenous ligase levels, as shown by quantitative biochemical analysis (Fig. 2e and Extended Data Fig. 2f). Although treatment of cells with the Cixu bivalent antibody was occasionally accompanied by IGF1R degradation, ligase-driven IGF1R degradation, particularly through ZNRF3, appeared to be significantly more robust (Fig. 2e). Limited or no degradation was observed in cell lines with no ligase expression, such as RKO cells, which do not have oncogenic Wnt pathway activation (Extended Data Fig. 2g). Consistently, IGF1R degradation was abolished in both RNF43-knockout (KO) and ZNRF3-KO HT29 cells that lack ligase cell-surface expression (Extended Data Fig. 2h,i). Together, these results demonstrate the ability of PROTABs to drive IGF1R degradation via endogenous ligase tethering.

With well-characterized IGF1R PROTABs in hand, we next explored some of their biological activities. First, we compared the effect of IGF1R degradation on downstream signalling events between PROTABs and ligand-competitive IgGs in SW48 cells. Cixu*ZNRF3 PROTABs inhibited IGF-driven phosphorylation of AKT and S6 more robustly than Cixu alone (Fig. 2f). Notably, decreased signalling also affected in vitro tumour cell viability, particularly at low antibody concentrations (Fig. 2g). Second, we assessed whether PROTABs could overcome the poor bioavailability and cell permeability that is frequently observed with PROTACs. PROTAB activity was evaluated following intravenous injection into mice bearing subcutaneously grafted SW48 tumours. A single (1 mg kg$^{-1}$) dose of ZNRF3*IGF1R PROTAB drove substantial degradation of IGF1R. Moreover, IGF1R degradation was partially suppressed at the highest dose (Extended Data Fig. 2j), suggestive of the hook effect reported with some PROTACs[22,23]. Notably, although IGF1R degradation was evident in implanted tumours, the PROTABs were well tolerated (Extended Data Fig. 2k) and no measurable IGF1R degradation was observed in the normal colonic mucosa (Fig. 2h, i). We also characterized PROTAB activity in human colon organoids, which more accurately recapitulate human physiology and disease, to validate the apparent tumour selectivity. Indeed, PROTABs could efficiently degrade IGF1R in multiple patient-derived CRC organoids while showing little to no activity in normal colon organoids. By contrast, bivalent Cixu showed indiscriminate degradation in both normal and cancer organoids (Fig. 2j). Collectively, these data illustrate that ZNRF3*IGF1R PROTABs are tumour-selective for CRC compared with the corresponding normal tissue.

## Characterization of ZNRF3*IGF1R PROTABs

Having demonstrated PROTAB activity both in vitro and in vivo, we sought to characterize the underlying mechanism of action. Copy number analysis of ZNRF3 and IGF1R across several cell lines revealed that the ratio of IGF1R:ZNRF3 was greater than 400:1 for some lines. Yet, efficient IGF1R clearance was still observed, implicating a catalytic rather than stoichiometric activity (Fig. 3a and Extended Data Fig. 3a,b). To probe this further, we focused on ZNRF3 in HT29 cells, which are readily amenable to CRISPR–Cas9 gene editing. We introduced insertion–deletion mutations (indels) to truncate ZNRF3 at the N-terminal domain (ZNRF3 N-term(i)) or RING domain (ZNRF3 RING(i)), which is required for ubiquitination (Extended Data Fig. 3c,d). Whereas ZNRF3 N-term(i) resulted in a complete loss of ligase expression, ZNRF3 RING(i) increased cell-surface expression compared with wild-type ZNRF3 (Fig. 3b), probably owing to defective auto-ubiquitination. Of note, PROTAB-mediated IGF1R degradation was abolished in both cell lines compared with parental cells (Fig. 3c). Cells ectopically expressing a RING deletion mutant (ZNRF3(ΔRING)) were generated and treated with PROTABs. In contrast

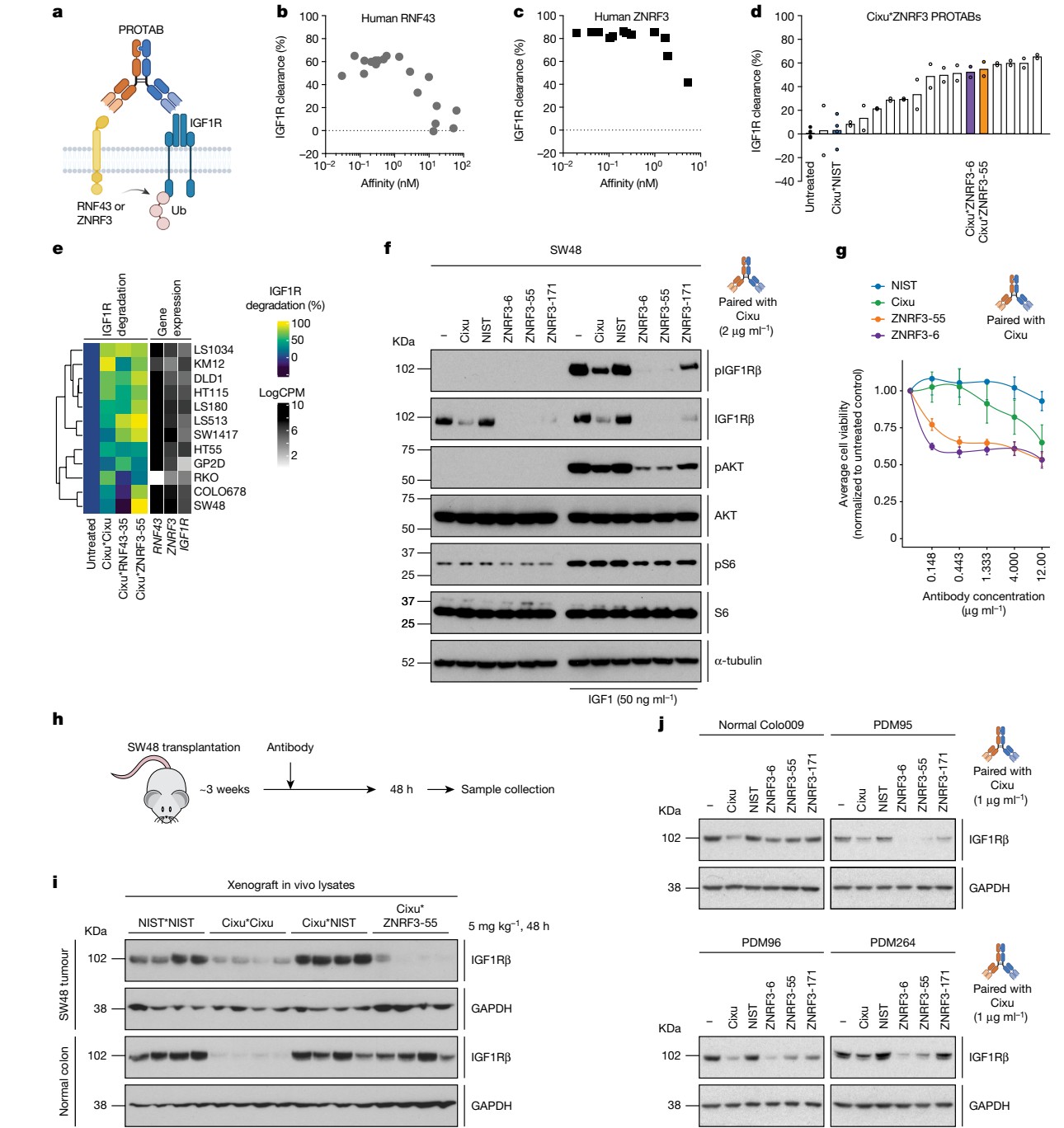

**Fig. 2 | Tethering endogenous RNF43 or ZNRF3 to IGF1R induces target internalization and degradation. a**, Schematic representation of ligase-based PROTAB-mediated IGF1R degradation. **b,c**, Levels of cell-surface IGF1R assessed by flow cytometry following treatment of HT29 cells expressing gD–RNF43–Flag (**b**) or gD–ZNRF3–Flag (**c**) with campaign PROTABs for 24 h. Graphs depict IGF1R clearance versus bivalent affinities of ligase campaign antibodies from one high-throughput FACS screening campaign. **d**, Levels of cell-surface IGF1R following treatment of HT29 HiBiT-IGF1R knock-in (KI) cells with ZNRF3*IGF1R bispecific PROTABs for 24 h. The graph depicts IGF1R clearance from one screening campaign with values from technical repeats overlayed. Assay controls and PROTABs used subsequently are highlighted. **e**, Levels of IGF1R-β following treatment of various CRC cell lines with indicated antibodies for 24 h. Per cent degradation is normalized to Cixu*NIST treatment and is the average of two independent experiments. Expression levels of key

genes are indicated. CPM, counts per minute. **f**, Levels of total and phosphorylated IGF1Rβ, AKT and S6 following treatment of SW48 cells with 2 µg ml⁻¹ of the indicated antibodies for 24 h with or without IGF1 stimulation for 20 min. α-Tubulin was used as a loading control. Data are representative of three independent experiments. **g**, Viability of SW48 cells following treatment with indicated antibodies for six days. Data are mean ± s.e.m. from three independent experiments. **h**, Schematic representation of SW48 xenograft in vivo model. **i**, Levels of IGF1Rβ in SW48 xenografts and normal colon tissues 48 h after intraperitoneal administration of 1 µg ml⁻¹ of indicated antibodies. GAPDH was used as a loading control. n = 4 animals per group. **j**, Levels of IGF1Rβ following treatment of normal or tumour-derived organoids treated with 1 µg ml⁻¹ of indicated antibodies for 24 h. GAPDH was used as a loading control. Data are representative of two independent experiments. For gel source data, see Supplementary Fig. 1.

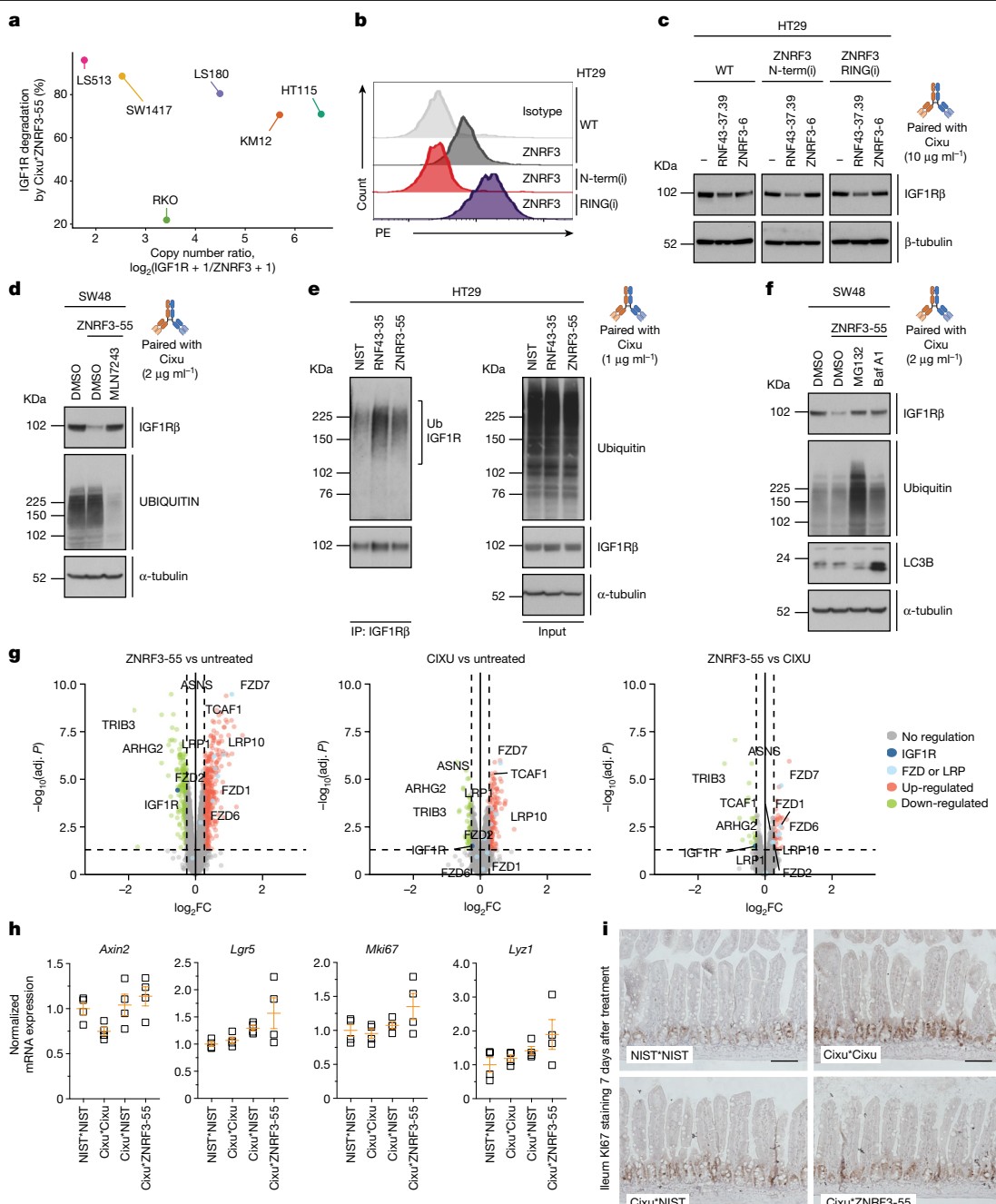

**Fig. 3 | ZNRF3\*IGF1R bispecific PROTABs induce target degradation in a ligase-dependent manner. a**, Clearance of IGF1R in cells treated with PROTABs (Fig. 2e) versus the basal IGF1R:ZNRF3 cell-surface ratio determined by copy number analysis (Extended Data Fig. 3a,b). **b**, Levels of cell-surface ZNRF3 in indicated cell lines. Data are representative of two independent experiments. **c**, Levels of IGF1Rβ in HT29 cells endogenously expressing wild-type or indicated ZNRF3 indels following treatment with 10 μg ml⁻¹ of the indicated PROTABs for 24 h. β-Tubulin was used as a loading control. Data are representative of two independent experiments. **d**, Levels of IGF1Rβ and ubiquitin in SW48 cells treated with DMSO or E1 inhibitor MLN7243 for 2 h followed by PROTAB treatment for 6 h. Data are representative of three independent experiments. **e**, Levels of total and immunoprecipitated IGF1Rβ and ubiquitin in HT29 cells subjected to denaturing IGF1Rβ immunoprecipitation (IP) following treatment with indicated antibodies for 2 h. Data are representative of two independent experiments. **f**, Levels of IGF1Rβ, ubiquitin and LC3B in SW48 cells treated with DMSO, the proteasome inhibitor MG132 or the lysosomal pathway inhibitor bafilomycin A1 (BafA1) for 2 h followed by PROTAB treatment for 6 h. Data are representative of three independent experiments. In **d**–**f**, α-tubulin was used as a loading control. **g**, Quantitative proteomic analysis of SW48 cells treated with indicated antibodies for 24 h. Volcano plots depict the average abundance fold change of 8,173 proteins from four independent experiments. **h**, Levels of Wnt-related gene transcripts in intestinal tissue from mice treated with indicated antibodies for 24 h. Data are mean ± s.e.m. with values from individual mice overlaid. *n* = 4 mice per group. **i**, Intestinal architecture of mice treated with indicated antibodies for 7 days. *n* = 4 mice per group. Scale bars, 100 μm. For gel source data, see Supplementary Fig. 1.

to wild-type ZNRF3, less efficient IGF1R degradation was observed in ZNRF3(ΔRING)-expressing cells (Extended Data Fig. 3e), with residual activity potentially attributed to endogenous ligase. LRP6, a putative substrate of ZNRF3, co-precipitated with both wild-type and ΔRING

proteins (Extended Data Fig. 3f), indicating that deletion of the RING domain does not alter protein integrity. Together, these results suggest that optimal PROTAB activity is dependent on the RING and C-terminal domains of ZNRF3. We further validated the catalytic dependency in

SW48 cells. These cells respond to PROTAB treatment within 6 h (data not shown), a time point at which small molecule inhibitors of the proteasome–lysosome machinery show minimal toxicity and off-target effects. Treatment of SW48 with MLN7243, a small molecule inhibitor of E1 ubiquitin ligases, impaired ZNRF3-mediated IGF1R degradation (Fig. 3d), implicating ZNRF3 catalytic function in PROTAB-mediated IGF1R degradation. Moreover, PROTAB treatment resulted in direct IGF1R ubiquitination in HT29 and DLD1 cell lines (Fig. 3e and Extended Data Fig. 3g). Collectively, our data demonstrate that PROTABs tethering ZNRF3 and IGF1R function by inducing ZNRF3-mediated IGF1R ubiquitination and degradation.

To investigate the route of target degradation, we next examined the effect of the proteasome inhibitor MG132 and the autophagic–lysosomal pathway inhibitor bafilomycin A1 on PROTAB activity. Similar to the canonical ZNRF3 substrates, FZD and LRP, ZNRF3-induced IGF1R degradation is mediated via the lysosomal pathway[10]. Notably, we also observed impaired IGF1R degradation upon proteasome inhibition with MG132 treatment of short duration, in which lysosomal dysfunction, as monitored by LC3B lipidation, is minimal (Fig. 3f). This suggests that PROTABs can engage both degradative pathways.

Next, we evaluated degradation specificity in response to PROTAB treatment across the proteome using quantitative mass spectrometry. The tested ZNRF3*IGF1R PROTAB significantly reduced IGF1R levels, whereas the bivalent Cixu antibody only modestly affected IGF1R levels (Extended Data Fig. 3h). In addition to changes in IGF1R levels, several proteins also exhibited down-regulation after treatment with either Cixu or the ZNRF3*IGF1R PROTAB and thus were probably a downstream consequence of IGF1R pathway inhibition (Fig. 3g and 'Data availability' section). Notably, peptides associated with endogenous ZNRF3 substrates, including LRP and FZD family members, increased in abundance with PROTAB treatment (Fig. 3g and Extended Data Fig. 3h). Collectively, the proteomics data validate the specificity and mechanism of action of PROTABs through concomitantly revealing the stabilization of endogenous ZNRF3 substrates with the decrease of the specific PROTAB-targeted receptor.

The observation that PROTAB treatment stabilized FZD and LRP proteins prompted the assessment of their effect on Wnt signalling activity. ZNRF3*IGF1R PROTAB treatment led to a modest increase in Wnt activity compared with validated Wnt agonists (Extended Data Fig. 3i). In vivo PROTAB treatment caused a minor increase in Wnt signalling in the gut, as evidenced by levels of known Wnt-responsive genes (Fig. 3h); however, no discernible effect on intestinal architecture was observed upon prolonged exposure (Fig. 3i). Combined, these data demonstrate the limited effect of the tested ZNRF3*IGF1R PROTAB on Wnt signalling agonism and intestinal homeostasis.

## Expansion of PROTAB-addressable targets

To demonstrate the generality of the PROTAB platform, we targeted additional membrane-associated proteins for degradation. Human epidermal growth factor receptor 2 (HER2), a validated therapeutic target in breast and gastric cancers, has been implicated in resistance to anti-epidermal growth factor (EGFR) therapies in CRC[24]. A ZNRF3*HER2 PROTAB led to significant degradation of HER2 in SW48 cells (Extended Data Fig. 4a) and in subcutaneously implanted tumours (Fig. 4a). Notably, treatment of normal and patient-derived CRC organoids with the ZNRF3*HER2 PROTAB led to tumour-specific degradation, whereas anti-HER2 bivalent antibody resulted in degradation in both normal cells and tumour cells (Extended Data Fig. 4b).

Next, we generated PROTABs targeting programmed death-ligand 1 (PD-L1), which drives cancer cell immune evasion. Whereas bivalent anti-PD-L1 did not have a substantial effect on PD-L1 levels in vitro or in vivo, treatment with ZNRF3*PD-L1 PROTABs resulted in near-complete degradation of the receptor (Fig. 4b and Extended Data Fig. 4c). Similar to IGF1R, PD-L1 degradation was dependent on

ZNRF3 catalytic activity and progressed via both the proteasomal and lysosomal pathways (Extended Data Fig. 4d,e). Together, these results demonstrate the potential of using the PROTAB technology to target multiple therapeutically relevant cell-surface proteins.

## Expansion of addressable ligases

Cytosolic ligase-based degrader platforms exploit a limited number of E3 ubiquitin ligases, partly owing to the challenge of obtaining high-affinity ligands[23]. We sought to explore whether PROTABs could utilize a broader range of cell-surface E3 ubiquitin ligases. Of the 449 known E3 ubiquitin ligases, we identified 38 putative cell-surface E3 ubiquitin ligases defined by the presence of a signal peptide, transmembrane domains and/or predicted or known localization to the plasma membrane (Extended Data Fig. 5a,b). Focusing on ligases that shared sequence, structural and domain architecture similarity with RNF43 and ZNRF3, we generated HT29 inducible cell lines expressing N-terminal gD and C-terminal Flag-tagged ligases, and assessed their cell-surface expression by flow cytometry. Similar to gD-tagged RNF43 and ZNRF3, a number of the newly identified ligases were detected at the cell surface using an anti-gD antibody (Fig. 4c). To assess whether non-natural targets could be degraded by this set of cell-surface ligases, we treated HT29 gD–ligase–Flag cells with gD*IGF1R PROTABs and monitored IGF1R levels. Western blot analysis confirmed that colocalizing validated cell-surface E3 ubiquitin ligases to IGF1R drives degradation (Fig. 4d). Conversely, ligases that could not be detected on the cell surface by flow cytometry did not induce target degradation (Extended Data Fig. 5c). The portability of the PROTABs platform was further assessed by exploring degradation of HER2, PD-L1 and FZD5, three therapeutically relevant cancer targets encompassing membrane receptors of various families (receptor tyrosine kinases, immunoglobulins and G-protein-coupled receptors, respectively). Similar to IGF1R, PROTABs targeting the gD epitope could drive clearance of the targeted receptors (Fig. 4e and Extended Data Fig. 5d,e). Of note, several of the newly identified cell-surface E3 ubiquitin ligases exhibited discrete tissue expression patterns (Extended Data Fig. 5f), suggesting that the tissue-specific degradation seen with ZNRF3*IGF1R PROTABs could be applicable to other ligases. Combined, our data highlight the scope of the PROTAB platform for repurposing multiple cell-surface E3 ubiquitin ligases as on-demand degraders of plasma membrane proteins, thus making PROTABs potentially broadly applicable to various therapeutic areas.

## Optimization of the PROTAB platform

Having demonstrated the expandability of this platform to multiple cell-surface ligases and targets, we investigated whether we could enhance target degradation by protein engineering. We focused on increasing the activities of the less active RNF43*IGF1R PROTABs (Fig. 4f). The effect of valency was explored using 2+1 Fab–IgGs (Fig. 4g). In HT29 cells with ectopic ligase expression, binding to two copies of IGF1R was minimally differentiated from the standard bispecific PROTAB (Fig. 4g). Notably, binding two copies of the ligase did enhance clearance; however, it also reduced the level of cell-surface RNF43, presumably by inducing auto-ubiquitination in *trans* (Fig. 4h). Next, one-armed Fv–IgG formats were engineered to test the hypothesis that the distance between bound antigens could affect the kinetics of ubiquitin transfer and target degradation. IGF1R–RNF43 Fv–IgG drove higher clearance in HT29 cells with both ectopic and endogenous ligase expression (Fig. 4i and Extended Data Fig. 5g), whereas reversing the orientation resulted in loss of affinity to IGF1R and did not significantly improve clearance (Fig. 4i and data not shown). The Fv–IgG enhanced initial rates of IGF1R degradation (Extended Data Fig. 5h) and maximum degradation compared with the standard one target plus one ligase bispecific PROTAB (Fig. 4j and Extended Data Fig. 5i). Thus, careful

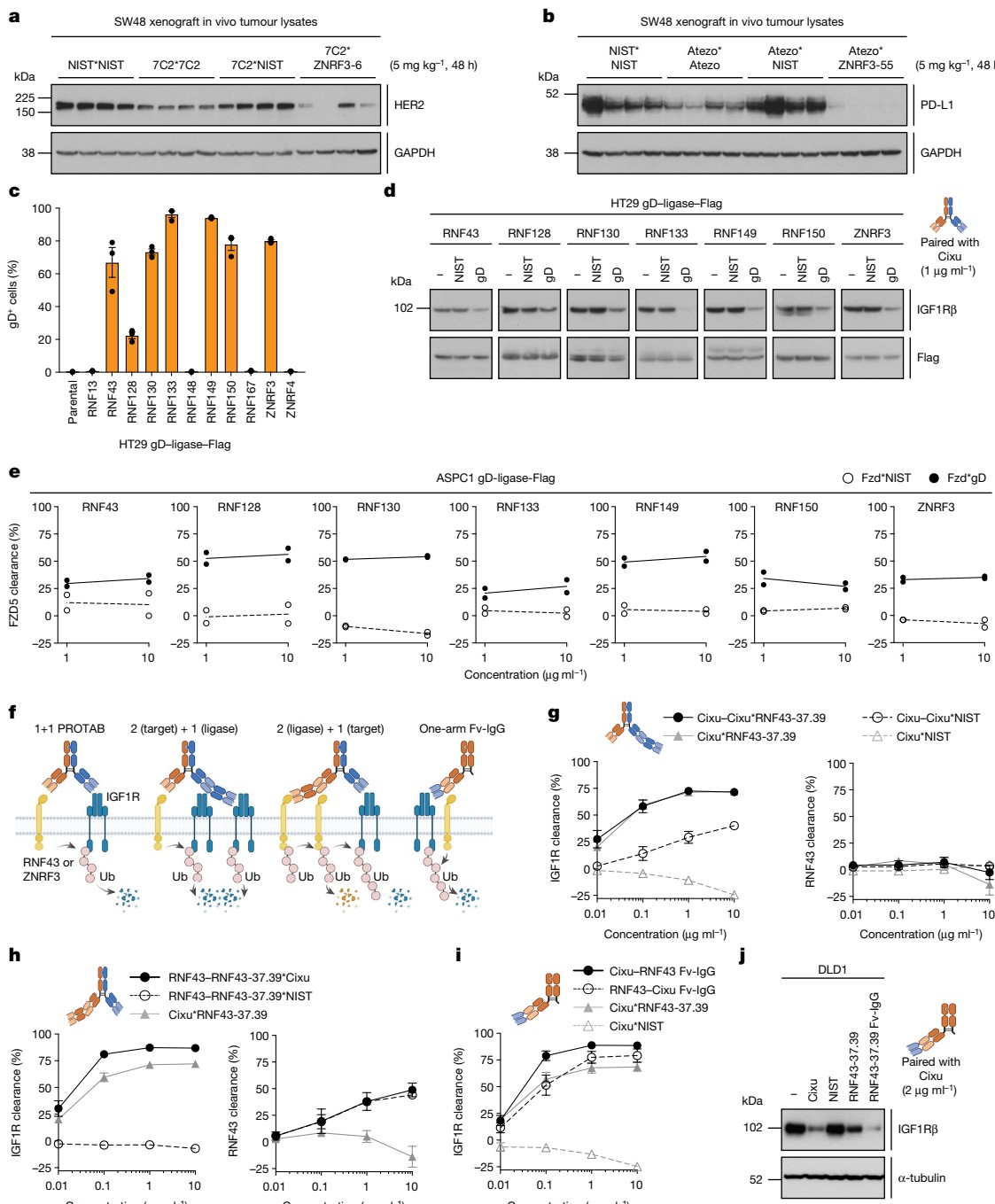

**Fig. 4 | Multiple cell-surface E3 ubiquitin ligases can degrade plasma membrane targets. a,b,** Levels of HER2 (**a**) and PD-L1 (**b**) in SW48 xenografts 48 h after intraperitoneal administration with indicated antibodies. GAPDH was used as a loading control. $n$ = 4 mice per group. **c,** Frequency of detectable cell-surface gD epitopes in HT29 cells expressing indicated gD–ligase–Flag constructs versus parental cells. Graph depicts gD-positive cells from three independent experiments. Data are mean ± s.e.m. with values from biological repeats overlaid. **d,** Levels of IGF1Rβ in HT29 cells expressing indicated gD–ligase–Flag constructs following treatment with specified antibodies for 24 h. Data are representative of three independent experiments. **e,** Levels of cell-surface FZD5 in ASPC1 cells expressing indicated gD–ligase–Flag constructs following antibody treatment at 1 or 10 µg ml⁻¹ for 24 h. Data are presented as lines connecting mean values from two independent experiments with values from biological repeats overlaid. **f,** Schematic representation of

PROTAB antibody engineering and the effect on downstream degradation. **g–i,** Levels of cell-surface IGF1R and RNF43 assessed by flow cytometry in HT29 cells expressing gD–RNF43–Flag following treatment with various antibody formats for 24 h: 2 (target) + 1 (ligase) (**g**), 2 (ligase) + 1 (target) (**h**), or one-arm Fv–IgG (**i**). The 2 (target) + 1 (ligase) and 2 (ligase) + 1 (target) antibodies are referred to as '2+1 Fab–IgGs'. Graphs depict IGF1R or RNF43 clearance from three independent experiments. Data are mean ± s.e.m. For **g–i,** samples were processed simultaneously using Cixu*NIST and Cixu*RNF43-37.39 as common controls for two (**g,i**) or three (**g,h**) biological repeats. **j,** Levels of IGF1R in DLD1 cells treated with indicated antibodies for 24 h. α-Tubulin was used as a loading control. Data are representative of two independent experiments. For gel source data and molecular masses of gD–ligase–Flag proteins in **d,** see Supplementary Fig. 1.

optimization of antibody affinity, format and epitope is central to enable efficient degradation of many targets of interest.

In summary, we outline the development of a platform that enables antibody-induced repurposing of cell-surface E3 ubiquitin ligases to drive efficient ubiquitination and degradation of transmembrane proteins. Indeed, during the course of this study a similar approach was reported in which RNF43-based bispecific antibodies (AbTACs) were used to degrade PD-L1[25]. Our work shows that PROTAB-mediated degradation is functionally differentiated from direct target inhibition, translates readily to in vivo systems, and can drive tumour-specific degradation. PROTABs can be broadly applied to transmembrane receptors and surface E3 ubiquitin ligases, potentially enabling ligase selection for tissue-specific target degradation. Although the full therapeutic potential of the PROTAB platform remains to be established, we anticipate that this technology, coupled with modular antibody engineering will provide additional opportunities to influence membrane receptor function with applications in both fundamental research and potential therapeutic discovery.

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

## Methods

### Cell lines

HEK293T, LS1034, KM12, DLD1, HT115, LS180, LS513, SW1417, HT55, GP2D, RKO, COLO678, SW48 and ASPC1 cell lines were obtained, characterized and controlled for quality as described[26]. Cell lines were maintained using standard tissue culture techniques. All cell lines were cultured in DMEM (HEK293T) or RPMI with 10% heat-inactivated fetal bovine serum, 2 mM L-glutamine and 1% penicillin/streptomycin. Mycoplasma negativity was ensured routinely.

### Organoid generation, culture and genome editing

Colon from adult C57BL/6 mice was removed, flushed, opened lengthwise and washed in cold PBS to remove all luminal contents. The colon was cut into 0.5–1 cm pieces in cold PBS, vortexed, washed for 3 times and placed in 25 ml 2.5 mM EDTA-PBS for 5 min at 37 °C. The supernatant was then removed and the colon pieces were washed with PBS followed by the incubation in 25 ml 5 mM EDTA-PBS for 15 min at 37 °C. After being vigorously vortexed, supernatant was collected and was filtered through 100-µm filters, spun at $500g$ for 5 min, washed with cold PBS and then resuspended 1:1 in Intesticult media (StemCell, 06005) and Matrigel (Corning, 356231). Intesticult media was changed every 2–3 days and organoids were passaged by mechanical disruption every 7–10 days.

To generate AKPS organoids, mutations in *Apc*, *Kras*[G12D] *Trp53* and *Smad4* were introduced by CRISPR–Cas9 technology. The following gene-specific single guide RNAs (sgRNAs) were used: *Apc*, CAGG ACTGCATTCTCCTGAA, AATGCAGTCCTGTCCCCATG, TTCTTGGG AATGACCCCATG; *Kras*, CTGAATTAGCTGTATCGTCA, G12D donor sequence; *Trp53*, GGAGCTCCTGACACTCGGAG; and *Smad4*, GATGTG TCATAGACAAGGTG. Organoids were dissociated into a single-cell suspension using accutase (Sigma-Aldrich) for 5 min at 37 °C and then electroporated with 2 µl of Cas9 and 3 µl of sgRNA using the P1 buffer and CM137 program (Lonza). After electroporation cells were embedded in Matrigel and DMEM advanced media (GIBCO) supplemented with 10 mM Hepes (Sigma-Aldrich), 2 mM GlutaMAX (Life Technologies), 1× Penicillin/Streptomycin (Life Technologies), 1× N2 (GIBCO), 1× B27 (GIBCO), 1 mM *N*-acetysteine (Sigma-Aldrich), 50 ng ml⁻¹ EGF (Life Technologies), 100 ng ml⁻¹ Noggin (Peprotech), and R-spondin-1 (R&D systems). Selection for mutated cells was performed using growth factor depletion from the culture medium: R-spondin depletion to select for *Apc*-deficient cells, EGF depletion to select for *Kras*⁺/G12D cells, presence of 10 µM Nutlin-3 (Sigma-Aldrich) to select for *Trp53* mutants, and Noggin depletion with presence of TGF-β (Peprotech, 10 ng ml⁻¹) to select for cells with *Smad4* deficiency.

Normal colon organoids were obtained from donor west and were cultured in Intesticult Organoid Growth Medium (StemCell, 06010). Patient-derived CRC organoids PDM95, PDM96 and PDM2674 were purchased from ATCC, HCM-CSHL-0142-C18, HCM-CSHL-0143-C20 and HCM-CSHL-0382-C19. Established CRC organoids were cultured in ATCC organoid media #1. Organoid media #1 consists of Advanced DMEM:F12 (Thermo Fisher, 12634028), 10 mM HEPES (Genentech), 2 mM L-glutamine (ATCC, 30–2214), 1× B27 (Thermo Fisher, 17504-044), 100 ng ml⁻¹ Noggin (Bio-techne, 6057-NG), 50 ng ml⁻¹ EGF (Bio-techne, 236-EG), 10 nM Gastrin (Bio-techne, 3006), 10 µM SB202190 (Bio-techne, 1264), 500 nM A83-01 (Bio-techne, 2939), 10 mM nicotinamide (LKT Labs, N3310) and 1.25 mM *N*-acetylcysteine (LKT Labs, A0918).

All human colon organoids were grown in 50% Matrigel plugs (Corning, 56231). At confluency, media was aspirated from well and organoid plugs were resuspended in PBS with pipetting to dissociate extracellular matrix. Cells were centrifuged at $500g$ for 5 min at 4 °C before aspiration of PBS and resuspension of cells in TrypLE Express (Gibco, 12604013). Organoids were dissociated by incubation at 37 °C for 3–5 min with intermittent pipetting to aid dissociation. After enzymatic and mechanical dissociation, cold basal media was added to organoids and cells were centrifuged at $500g$ for 5 min at 4 °C. Media

was carefully aspirated, and subsequently organoids were washed one time with cold PBS and centrifuged at $500g$ for 5 min at 4 °C. PBS was carefully aspirated and organoids were resuspended in 50:50 mix of complete media and Matrigel. Organoids were seeded in 3D domes on pre-warmed plates and allowed to solidify for 5 min at 37 °C before plate inversion and 15 min before addition of media.

### In situ hybridization and immunostaining

Hybridizations using the RNAscope method were performed according to the manufacturer's protocol (Advanced Cell Diagnostics) using the RNAscope 2.5 HD Reagent Kit-RED (322350). Probes used were Mm*Zrnf3* (434201), Mm*Rnf43* (400371).

For immunohistochemistry, samples were formalin-fixed and paraffin-embedded using standard procedures. For Ki67 antibody stains, rehydrated sections were pressure-cooked for 15 min in antigen unmasking buffer (DAKO), blocked in serum-free protein block (DAKO) and incubated in anti-KI67 (Sigma-Aldrich, 1:400) overnight. Sections were then incubated in HRP-conjugated anti-rabbit antibody (DAKO, 1:200) and detected with DAB reaction (DAKO). Pictures were acquired with a NIKON A1R using N3 elements software.

### Gene expression analysis

RNA sequencing (RNA-seq) libraries were prepared with the TruSeq RNA Sample Preparation kit (Illumina, CA). The libraries were sequenced on Illumina HiSeq 2500 sequencers to obtain on average 34 million 50-bp single-end reads per sample. RNA-seq reads were first aligned to ribosomal RNA sequences to remove ribosomal reads. The remaining reads were aligned to the mouse reference genome (NCBI Build 38) using GSNAP37 version 2013-10-10, allowing a maximum of two mismatches per 50-base sequence (parameters: -M 2 -n 10 -B 2 -i 1 -N 1 -w 200000 -E 1 –pairmax-rna = 200000 –clip-overlap). Transcript annotation was based on the RefSeq database (NCBI annotation release 104). To quantify gene expression levels, the number of reads mapped to the exons of each RefSeq gene was calculated. Read counts were scaled by library size, quantile normalized and precision weights were calculated using the voom R package. Subsequently, differential expression analysis on the normalized count data was performed using the limma R package[27]. In addition, gene expression was obtained in the form of normalized reads per kilobase gene model per million total reads (nRPKM) as described[28].

For quantitative PCR (qPCR), RNA was isolated from samples using either the RNeasy Micro or Mini kit (Qiagen, 74004 and 74104). qPCR was performed in 10 µl reactions with 50 ng total RNA using the One-step Real-time RT–PCR mastermix (Life Technologies, 4392938) according to the manufacturer's instructions. Taqman probes *Axin2* (Mm00443610_m1), *Lgr5* (Mm00438890_m1), *Mki67* (Mm01278617_m1) and *Lyz1* (Mm00657323_m1) were from Life Technologies. qPCR reactions were run on a 7900HT Fast Real-Time PCR system (ABI) at the following thermal cycling conditions: holding step of 30 min at 48 °C followed by a holding step of 10 min at 95 °C and 40 cycles of 10 s at 95 °C and 1 min at 60 °C.

### Animal studies

Wild-type B57BL/6 mice (000664) and NOD.Cg-PrkdcscidIl2rgtm1Wjl/ SzJ (NSG) (colony 005557) mice were purchased from the Jackson Laboratory. Females of 6 to 12 week old were used for experiments. Female 8-to-10-week-old Sprague Dawley rats were obtained from Charles River. Female 16-week-old white rabbits were obtained from WORC (Western Oregon Rabbit). Animal studies were approved by Genentech's Institutional Animal Care and Use Committee and adhere to the NRC Guidelines for the Care and Use of Laboratory Animals.

### In vivo engraftment studies

Approximately, 1 million SW48 cells were resuspended in PBS basal media, admixed with 50% Matrigel (Corning) to a final volume of 200 µl,

and injected subcutaneously in the left flank of NSG mice. Tumour dimensions were measured using calipers and tumour volume was calculated as $0.523 \times$ length $\times$ width $\times$ width. Mice were humanely euthanized according to the following criteria: clinical signs of persistent distress or pain, significant bodyweight loss (>20%), tumour size exceeding 2,500 mm$^3$, or when tumours ulcerated. Maximum tumour size permitted by the Institutional Animal Care and Use Committee (IACUC) is 3,000 mm$^3$ and in none of the experiments was this limit exceeded. When tumours reached ~400 mm$^3$, mice were randomized to receive a single intraperitoneal injection of the control antibodies or PROTABs. None of the experiments were blinded. All mice were euthanized 48–72 h after injection and tumours were collected for further processing. The lumen implantation procedure has previously been described[15]. In brief, mice were anaesthetized by isoflurane inhalation and injected intraperitoneally with buprenorphine at 0.05 to 0.1 mg kg$^{-1}$. A blunt-ended haemostat (Micro-Mosquito, 13010-12, Fine Science Tools) was inserted ~1 cm into the anus. The haemostat was angled toward the mucosa and opened slightly such that a single mucosal fold could be clasped by closing the haemostat to the first notch. The haemostat was retracted from the anus, exposing the clasped exteriorized mucosa. A 10 µl of solution containing 50,000 cells admixed with 50% Matrigel (Corning) in PBS was directly injected into the colonic mucosae. After reversing the prolapse, the haemostat was then released.

## Animal immunization

Antibodies were identified by immunizing rats or rabbits with recombinant RNF43 and/or ZNRF3. In brief, Sprague Dawley rats (Charles River) were immunized with a priming dose of 100 µg human (RNF43 or ZNRF3) protein solubilized in detergent mixed with MPL + TDM adjuvant (Sigma-Aldrich), CFA (Sigma-Aldrich) or mixed with a combination of TLR agonists: 50 µg MPL (Sigma-Aldrich), 20 µg R848 (Invivogen), 10 µg poly(I:C) (Invivogen), and 10 µg CpG (Invivogen) divided among multiple sites. New Zealand white rabbits were also immunized with a mixture of the same proteins solubilized in detergent mixed with CFA (Sigma-Aldrich). For additional protein boosts, the rats and rabbits received half the amount of the priming dose protein diluted in PBS. Rats and rabbits were dosed every two weeks. Polyclonal antisera from these rats and rabbits were purified and tested by ELISA for binding to human RNF43 and ZNRF3. For the rats, multiple lymph nodes were collected three days after the last immunization that showed detectable FACS reactivity against human RNF43 or ZNRF3. IgM-negative B cells from these rats were purified from whole lymphocytes using magnetic separation (Miltenyi Biotec) and stained with anti-rat IgM antibody (Jackson ImmunoResearch), anti-rat CD45RA (Biolegend), anti-rat CD8a (Biolegend), and labelled human RNF43-Alexa 633 or human ZNRF3-Alexa 633 by Lightning-Link Alexa 633 Antibody Labeling kit (Novus). Rat B cells showing minimal rat IgM expression while binding to the human RNF43 or ZNRF3 protein were sorted and deposited into 96-well plates containing a culture medium with feeder cells and supplemented with cytokines using a FACS Aria III sorter (BD). For the rabbits, the blood was collected three days after the last immunization that showed detectable FACS reactivity against human RNF43 and ZNRF3. IgG-positive B cells from these rabbits were purified from the whole blood using magnetic separation (Miltenyi Biotec) and stained with anti-rabbit IgG antibody (Southern Biotech) and labelled human RNF43-Alexa 633 or human ZNRF3-Alexa 633 by Lightning-Link Alexa 633 Antibody Labeling kit (Novus). Rabbit B cells showing maximum rabbit IgG expression while binding to the human RNF43 or ZNRF3 protein were sorted and deposited into 96-well plates containing a culture medium with feeder cells and supplemented with cytokines using a FACS Aria III sorter (BD). Supernatants were screened by ELISA against human RNF43 or ZNRF3 seven days after sorting. Supernatants demonstrating human RNF43 or ZNRF3 binding were tested by FACS for binding to human RNF43 or ZNRF3 expressed on the surface of gD–RNF43-expressing cells or binding to human ZNRF3 expressed on the surface of gD–ZNRF3-expressing cells. RNA was extracted from B cells that showed RNF43 or ZNRF3 FACS binding for molecular cloning and recombinant expression. Recombinant antibodies were tested by FACS for binding to human RNF43 expressed on the surface of gD–RNF43-expressing cells or binding to human ZNRF3 expressed on the surface of gD–ZNRF3-expressing cells.

## Antibody generation

DNA encoding antibody heavy and light-chain variable domains was generated by gene synthesis. The synthesized gene fragments were inserted into mammalian expression vectors containing the corresponding heavy or light constant domains. Species and isotypes included human IgG1 and murine IgG2a. Some variable domain sequences were edited to remove apparent unpaired cysteine residues and NX[S/T] N-glycosylation motifs. Recombinant antibodies were produced by transient transfection of Expi293 cells with mammalian expression vectors encoding the antibody heavy chain and light chain. Heavy chain and light chain were encoded on separate vectors, and were transfected using a 1:1 ratio of heavy chain expression vector to light chain expression vector. Antibodies were purified from the cell culture supernatant by affinity chromatography. In some cases, antibodies underwent an additional purification step based on SEC. Bispecific antibodies were generated using knob-into-hole technology[19] using human IgG1 or murine IgG2a backbones typically including mutations to reduce effector function (L234A, L235A and P329G). Antibodies containing either knobs or holes were expressed and purified prior to assembly into a bispecific format as previously described. In some cases, mutations were introduced to enable bispecific expression in a single cell including appropriate light-chain pairing.

Complex-format 2+1 target antibodies were designed as previously described and expressed as half-antibodies, using light-chain pairing mutations to generate the Fab–IgG arm, and then assembled as described above. One-armed Fv–IgGs were designed as fusions of $V_h$ or $V_l$ onto the heavy chain (HC) or light chain (LC), respectively. The HC, LC and a Fc-only arm, were co-expressed as described above using knob-into-hole technology to drive appropriate HC pairing.

## Epitope mapping

Antibodies were screened for binding to recombinant human RNF43 and ZNRF3 ECDs using a Biacore 8k instrument (GE Life Sciences). In brief, antibodies diluted to 1 µg ml$^{-1}$ in 1× HBSP buffer (Cytiva, BR100368) were captured on a Sensor Chip Protein A (GE Life Sciences) using a flow rate of 10 µl min$^{-1}$ and a contact time of 60 s. Binding of recombinant human RNF43 and ZNRF3 ECDs to the captured antibodies was analysed at 25 °C using a single-cycle kinetics method with a flow rate of 30 µl min$^{-1}$, a contact time of 180 s and a dissociation time of 600 s. The concentration of recombinant human RNF43 and ZNRF3 ECDs in the single-cycle kinetics were 0, 0.16, 0.80, 4, 20 and 100 nM. Between cycles, the chip was regenerated using 10 mM glycine HCl pH 1.5 injected for 30 s at 30 µl min$^{-1}$. Data were evaluated using Biacore 8K Evaluation software (GE Life Sciences). Kinetic constants were obtained using a 1:1 binding model with the parameter RI set to zero.

## High-throughput epitope binning

Antibodies generated by animal immunization were reformatted into hIgG1 backbones and binning was performed using the CFM2/MX96 surface plasmon resonance (SPR) system (Wasatch Microfluidics, now Carterra), equipped with DA v6.19.3, IBIS SUIT, SprintX and Carterra Epitope Tool software. Antibodies at 10 µg ml$^{-1}$ were immobilized on an SPR sensor prism CMD 200 M (Xantec Bioanalytics) by amine coupling using a 10 mM sodium acetate pH 4.5 immobilization buffer. Immobilization was performed with the CFM2 instrument and the sensor prism was then transferred to the IBIS MX96 instrument for SPR-based competition analysis. Immobilized antibodies were exposed

first to 100 nM recombinant human RNF43 or ZNRF3 ECDs and then to 10 µg ml$^{-1}$ antibody in solution, using an HBS-EP running buffer (10 mM HEPES, 150 mM NaCl, 0.05% Tween 20, pH 7.4, 1 mM EDTA). 10 mM glycine/HCl pH 1.7 was used as the regeneration buffer.

## Radiochemistry

Antibodies were indirectly radiolabelled with iodine-125 ($^{125}$I) using a modified indirect Chizzonite method as described previously[29]. In brief, $^{125}$I was obtained as sodium iodide in 0.1 N sodium hydroxide (Perkin Elmer). 1 mCi (~ 3 µl) of $^{125}$I was used to randomly label tyrosine residues at a specific activity of ~10 µCi µg$^{-1}$, following $^{125}$I activation with Iodogen tubes (Pierce Chemical).

## Copy number determination

The copy numbers of cell-surface IGF1R and ZNRF3 were determined by performing saturation binding experiments using the radiolabelled antibodies against IGF1R (cixutumumab) or ZNRF3 (ZNRF3–55). For saturation experiments, cells were incubated with increasing concentrations of $^{125}$I-labelled antibodies in binding buffer (OptiMEM, 2% FBS, 50 mM HEPES, and 0.1% sodium azide) for 12 h at room temperature under gentle agitation. Nonspecific binding was determined by pre-incubating cells with an excess of non-labelled antibodies prior to addition of $^{125}$I-labelled antibodies. Cells and antibodies were transferred to Millipore multiscreen filter plates, washed four times with binding buffer, and allowed to dry. The dried filters were punched into 5 ml polystyrene tubes (Corning) and the radioactivity in counts per minute (CPM) was measured using a Wallac WIZARD 2470 Gamma Counter (Perkin Elmer) for 1 min with 0.8 counting efficiency. Data were fit to a one-site specific binding curve in GraphPad Prism 8.

## CellTiter-Glo

Cells were seeded at a density of 2,500 cells per well in 96-well plates 24 h prior to treatment. Cells were treated with a dose titration of various antibodies. After 6 days of treatment, cell viability was measured using CellTiter-Glo Luminescent Cell Viability Assay (Promega, G7570) following manufacturer's protocol. Reagents and plates were equilibrated to room temperature.

## Wnt reporter assay

HEK293 cells transfected stably with a TOPbrite firefly luciferase Wnt reporter and pRL-SV40 *Renilla* luciferase (Promega, E2231) were maintained in RPMI 1640 media supplemented with 10% fetal bovine serum. For luciferase assay, cells were stimulated with CHIR99021 (StemCell Technologies, 72052), recombinant mouse WNT3A protein (R&D Systems, 1324-WN-002/CF) and the EC$_{50}$ of RSPO3 (10.5 pM, Genentech) or the various PROTABs at either 1 or 10 µg ml$^{-1}$. Following 16 h treatment, luciferase activity was detected using the Promega Dual-Glo system (Promega, E2920) according to the manufacturer's instructions. Data were analysed as the ratio of firefly/*Renilla* luciferase activity.

## HiBiT-LgBiT nano-luciferase assay

A HiBiT tag was introduced to the N-terminus of IGF1R by a CRISPR–Cas9 system in HT29 cells. Single clones with the highest NanoLuc luciferase luminescence readout were selected and verified by Sanger sequencing. Cells were plated onto 96-well flat clear bottom white polystyrene tissue culture-treated microplates at 100,000 cells per well and incubated in ATCC-recommended media in the presence of various concentrations of antibodies. After 24 h (or the indicated period of time), the cells were washed with 100 µl PBS once and then replenished with 100 µl fresh media. The detection reagent was prepared by diluting the LgBiT protein at a ratio of 1:100 and the substrate at a ratio of 1:50 into a desired volume of detection buffer supplied in the detection kit. 100 µl of the detection reagent was then added to the cells in 100 µl fresh media. For Nano-Glo HiBiT Extracellular Detection System, which detects the surface level of IGF1R, detection was performed upon 10 min incubation at room temperature with gentle mixing using a plate shaker. IGF1R clearance (%) was calculated using the relative light units (RLU) readings based on the following equation: percentage clearance = (untreated samples RLU − treated samples RLU)/(untreated samples RLU × 100%). RNF43 and ZNRF3 bispecific and multispecific PROTABs were assessed in this cell line. A subset of antibodies was run in two separate runs, in which case both results are included.

## Flow cytometry cell-surface clearance

HT29 or ASPC1 doxycycline-inducible gD–ligase–Flag cells were seeded at a density of 80,000–100,000 cells per well in 96-well plates for 24 h and treated with doxycycline for 24 h followed by antibodies. At various time points after treatment, cells were washed with PBS (100 µl per well), detached from the well surface by addition of Accutase (Millipore, 100 µl per well) at 37 °C for 10 min, and resuspended into single cells. Accutase activity was quenched by adding an equivalent volume of RPMI + 10% heat-inactivated Fetal bovine serum with penicillin, streptomycin, and glutamine. After centrifugation for 4 min at 1,200 rpm, the supernatant was discarded. Cells were resuspended in 150 µl FACS buffer (PBS, 0.5% BSA, 0.05% sodium azide) for 10 min on ice, centrifuged, and the supernatant was discarded. Cells were incubated with 40 µl of fluorescently-labelled staining antibodies (2–5 µg ml$^{-1}$ final concentration) for 45 min on ice. Staining antibodies include xIGF1R mIgG1-APC (1H7, ebioscience 17-8849-42), xRNF43 hSC37.17-APC (in-house generated reagent derived from WO2015164392A2 using the Alexa Fluor 647 Antibody labeling kit, Thermo Fisher, A20186). xFzd5 hIgG1 (in-house reagent, used at 5 µg ml$^{-1}$) was used as a primary staining antibody for 45 min on ice (5 µg ml$^{-1}$ final concentration), followed by two washes in FACS buffer (150 µl) and a secondary staining antibody Goat anti-Human IgG (H+L)-647 (Invitrogen, A21445, 1:1,000) for 45 min on ice. Background or isotype control staining antibodies include NISTMab hIgG1-APC and xRagweed mIgG1-APC (both in-house generated reagents). Viability dye-780 was added in conjunction with staining antibodies (eBioscience, 65-0865-14, 1:1,000). Post-staining, cells were washed twice in FACS buffer (150 µl) and resuspended in 40 µl FACS buffer for analysis.

Flow cytometry was performed on a BD FACSCelesta flow cytometer using the high-throughput sampler in high-throughput mode. Samples were resuspended twice, then analysed by the cytometer (10 µl sample volume, 180 µl min$^{-1}$ flow rate), which measured SSC, FSC and APC signals. Between each sample, the system was washed with 400 µl buffer. Using FlowJo FACS analysis software, single-cell gating was obtained using the SSC and FSC profiles. The average APC median fluorescent intensity (MFI) was used as a measure of cell-surface target expression, and per cent surface clearance was calculated as (1 − (treated MFI − background)/(untreated MFI − background)) × 100. Results were plotted in Prism 8.

## iDimerize assay

The effect of chemically induced dimerization of RNF43 or ZNRF3 with IGF1R was assessed using the iDimerize system (TakaraBio). In brief, HEK293T cells were reverse transfected with pBind plasmids to induce the constitutive co-expression of RNF43–DmrA–HA and IGF1R–DmrC–Flag or ZNRF3–DmrA–HA and IGF1R–DmrC–Flag. After 18–20 h, cells were either left untreated or treated with varying concentrations of the A/C heterodimerizer (TakaraBio, 635057) to induce ligase–target dimerization and cells were incubated for an additional 24 h. Cells were then lysed in ice-cold GST lysis buffer (25mM Tris•HCl, pH 7.2, 150 mM NaCl, 5 mM MgCl2, 1% NP-40 and 5% glycerol, 1% Halt protease and phosphatase inhibitor cocktail). Cleared lysates were subjected to an HA immunoprecipitation by incubating lysates with 25 µl Pierce HA epitope tag antibody agarose conjugate (2-2.2.14) (Thermo Fisher, 26182) for 3 h at 4 °C. For input samples, 20 µl were used from the prepared immunoprecipitation samples before incubation. Beads were washed following incubation and prepared for western blot analysis.

## Flag immunoprecipitation

HEK293T parental or doxycycline-inducible gD–ZNRF3-Flag cells were plated at a density of 5 million cells per 15 cm dish. Cells were subjected to media exchange with doxycycline treatment 24 h and 48 h post plating. Cells were then lysed in ice-cold GST lysis buffer. Cleared lysates were subjected to Flag immunoprecipitation by incubating lysates with 50 µl Flag M2 Affinity Gel (Sigma-Aldrich, A2220) for 150 min at 4 °C. For input samples, 60 µl were used from the prepared immunoprecipitation samples prior to incubation. Beads were washed following incubation and either directly prepared for western blot analysis or subjected to elution. To elute proteins bound to the beads, supernatant was aspirated using gel loading tips to avoid bead loss and resuspended in 400 µl Flag elusion buffer prepared by diluting 5 mg ml$^{-1}$ 3× Flag peptide stock solution (Sigma-Aldrich, F4799) in GST lysis buffer at a final concentration of 200 µg ml$^{-1}$ and incubated for 30 min at 4 °C. Following incubation samples were transferred to microspin columns (GE Healthcare, 27356501) to separate the eluate from the beads and spun at 6,000g for 30 s. Eluates were transferred to Amicon filter units with a 3K MWCO (Millipore, UFC500396) and spun down at 14,000g for 30 min at room temperature. Filters were then inverted into new tubes and spun down at 1,000g for 2 min at room temperature and concentrated samples were prepared for western blot analysis.

## IGF1R ubiquitination assay

IGF1R ubiquitination following antibody treatment was evaluated in HT29 and DLD1 cells. In brief, 40 million cells were plated in 15-cm dishes and incubated for 72 h. Cells were subjected to a media exchange concomitant with bivalent or bispecific antibody treatment (1 µg ml$^{-1}$). Cells were incubated for 2 h and 15 min at 37 °C. Following incubation, cells were scraped in 800 µl ice-cold PBS + 10 mM NEM, 1 mM PMSF, 1% Halt protease and phosphatase inhibitor cocktail and cell pellets were resuspended in 400 µl denaturing buffer (2% SDS, 150 mM NaCl, 10 mM Tris.HCl, pH 7.5, 10 mM NEM, 1 mM PMSF, 1% Halt protease and phosphatase inhibitor cocktail) and mixed by vortexing prior to 3 min incubation at 95 °C. Boiled samples were renatured by adding 800 µl NP-40 immunoprecipitation buffer (50 mM Tris•HCl, pH 7.5, 0.5% IGE-PAL, 150 mM NaCl, 10 mM NEM, 1 mM PMSF, 1% Halt protease & phosphatase inhibitor cocktail). Samples were sonicated twice for 32 s with 2 s on/off pulse intervals at 30% AMP on ice. Samples were centrifuged at 16,100 g for 15 min at 4 °C. Cleared lysates were incubated with 25 µl per sample prewashed Dynabeads protein G (Thermo Fisher, 10004D) conjugated with IGF1 receptor (D23H3) XP rabbit monoclonal antibody (Cell Signaling; 9750) for IGF1R immunoprecipitation at 1.87 µg per sample. Lysate-bead mixtures were incubated for 24 h at 4 °C. For input samples, 60 µl were used from the prepared immunoprecipitation samples prior to incubation. Beads were washed following incubation and prepared for western blot analysis.

## Western blot analysis

Various cell lines were plated in 6-well plates (Corning, 3516) at a density of 0.5–1 million cells per well and left to adhere overnight. Cells were treated with antibodies at specified concentrations and durations. Where applicable, cells were pre-treated with doxycycline (0.5–1 µg ml$^{-1}$), E1 inhibitor (MLN7243 1 µM), MG132 (Sigma-Aldrich, M7449, 10 µM) and bafilomycin A1 (Sigma-Aldrich, SML1661, 100 nM). Following incubation, cells were lysed in GST lysis buffer or RIPA buffer supplemented with 1% Halt protease and phosphatase inhibitor cocktail (Thermo Fisher, 78442) for 20 min on ice. Lysates were clarified and prepared for western blot analysis using NuPAGE LDS Sample Buffer (4×) (Invitrogen, NP0007) and NuPAGE LDS Sample Reducing Agent (10×) (Invitrogen, NP0009). Samples were run on 4–12% Bis-Tris gel (Invitrogen, WG1403) using BOLT or NuPAGE MOPS SDS running buffer at 90–100 V for approximately 2–3 h. For molecular weight determination samples were run alongside Amersham ECL Rainbow marker, full range (Cytiva, RPN800E) or equivalent marker. Gels were transferred to Nitrocellulose membranes using iBlot2 system using iBlot2 Nitrocellulose Regular stacks (Invitrogen, IB23001) and transferred using protocol P0 (1 min at 20 V, 4 min at 23 V, and 2 min at 25 V). Alternatively, gels were transferred using Immun-Blot PVDF/filter paper sandwiches (Bio-Rad, 1620239) at 100 V for 100 min. Blots were blocked with 5% milk in PBST/TBST for 1 h at room temperature. Blots were transferred to 5% milk in PBST/TBST containing antibodies outlined in the antibodies section and incubated at room temperature for 1 h or overnight at 4 °C. Blots were washed 3–4 times for 5 min each with PBST/TBST before incubation in 5% milk containing IRDye secondary antibody for 1 h at room temperature. Blots were washed 4 times for 5 min each and developed using the Li-Cor system. Blots developed with film were incubated with primary antibodies, washed 4 times for 5 min each with PBST/TBST before incubation in 5% milk containing anti-rabbit or anti-mouse HRP secondary antibody for 1 h at room temperature. Blots were washed 4 times for 5 min each and developed using film and blotting substrate (Thermo Fisher, 34075, used at 1:10) or the Amersham ECL Select western blotting detection reagent (Cytiva, RPN2235, used as supplied or at 1:10 and 1:20).

## Antibodies

Akt (pan) (40D4) mouse monoclonal antibody, Cell Signaling 2920S lot 8, dilution 1:1,000; anti-HA high affinity (3F10), Roche Diagnostics 11867423001 lot 45715900, dilution 1:1,000; anti-mouse IgG, HRP-linked antibody, Cell Signaling, 7076S lot 36, dilution 1:5,000; anti-rabbit IgG, HRP-linked antibody, Cell Signaling 7074S lot 30, dilution 1:5,000; anti-rat IgG, HRP-linked antibody, Cell Signaling 7077S lot 14, dilution 1:5,000; GAPDH (14C10) rabbit monoclonal antibody (HRP conjugate), Cell Signaling 3683 lot 4, dilution 1:1,000; goat anti-human IgG (H+L) secondary antibody, Alexa Fluor 647, Invitrogen A21445 lot 2339821, dilution 1:1,000; goat anti-mouse IgG (H+L) secondary antibody, HRP, Invitrogen 32430 lot VD301382, dilution 1:5,000; goat anti-rabbit IgG (H+L) secondary antibody, HRP, Invitrogen 32460 lot VE30198, dilution 1:5,000; HER2/ErbB2 (D8F12) XP rabbit monoclonal antibody, Cell Signaling 4290S lot 6, dilution 1:1,000; IGF-I receptor β (D23H3) XP rabbit monoclonal antibody, Cell Signaling 9750S lot 5–7, dilution 1:1,000; IGF1R (1H7), APC, eBioscience 17-8849-42 lot 2330467, dilution 1:20; IRDye 800CW donkey anti-mouse IgG secondary antibody, LI-COR 926–32212 lot D10414-15 and D00930-09, dilution 1:5,000; IRDye 800CW goat anti-rabbit IgG secondary antibody, LI-COR 926–32211 lot D10629-12 and D01110-10, dilution 1:5,000; LC3B antibody, Novus Biologicals NB100–2220 lot EU/EU-3, dilution 1:1,000; LIVE/DEAD Fixable Violet Dead Cell Stain Kit, for 405 nm excitation, Thermo Fisher L34964 lot 2208471, dilution 1:1,000; LRP6 (C5C7) rabbit monoclonal antibody, Cell Signaling 2560S lot 11, dilution 1:1,000; monoclonal anti-Flag M2 antibody produced in mouse, Sigma-Aldrich F3165 lot SLCG2330 and SLBT6752, dilution 1:1,000; monoclonal anti-α-tubulin antibody produced in mouse, Sigma-Aldrich T9026 lot 099M4773V, dilution 1:10,000; PD-L1 (E1L3N) XP rabbit monoclonal antibody, Cell Signaling 13684S lot 18, dilution 1:1,000; phospho-AKT (Ser473) antibody, Cell Signaling 9271S lot 15, dilution 1:1,000; phospho-IGF-I receptor β (Tyr1135/1136)/insulin receptor β (Tyr1150/1151) (19H7) rabbit monoclonal antibody, Cell Signaling 3024S lot 11, dilution 1:1,000; phospho-S6 ribosomal protein (Ser235/236) antibody, Cell Signaling 2211S lot 23, dilution 1:1,000; polyclonal rabbit anti-human c-ErbB-2 cncoprotein, Agilent (DAKO) A0485 lot 20083958, dilution 1:1,000; S6 ribosomal protein (5G10) rabbit monoclonal antibody, Cell Signaling 2217S lot 10, dilution 1:1,000; ubiquitin monoclonal antibody (P4G7-H11), Enzo ADI-SPA-203-F lot 04062139, dilution 1:1,000; Ultra-LEAF purified human IgG1 isotype control recombinant antibody, BioLegend 403502 lot B322065, dilution 10 µg ml$^{-1}$; vinculin antibody, Cell Signaling 4650S lot 5, dilution 1:1,000; β-tubulin (9F3) rabbit monoclonal antibody, Cell Signaling 2128L lot 11, dilution 1:1,000.

## Mass spectrometry

SW48 cells were plated at 40 million cells per 15-cm dish for 24 h. Cells were rinsed once in ice-cold PBS and collected by scraping in 1 ml ice-cold PBS following 24 h treatment with specified antibodies. Pellets were collected by centrifugation and processed in two independent sets. The pellets were lysed in 9M urea lysis buffer (9M urea, 1 mM sodium orthovanadate, 2.5 mM sodium pyrophosphate, 1 mM β-glycerophosphate, 20 mM HEPES, pH 8.0). The lysis buffer containing pellets were sonicated using a microtip at 20 W output for 5 s for 3 bursts before centrifugation at 16,000$g$ for 10 min at room temperature. The supernatant was collected and protein quantitation was performed using BCA assay (Pierce, 23225). Approximately, 500 µg total protein amount was used per sample for the reduction, alkylation, and digestion. The first sample set was processed by diluting 6 M guanidine to 2 M with 100 mM 3-[4-(2-hydroxyethyl)-1-piperazinyl] propanesulfonic acid (EPPS), pH 8.0. Approximately 3 mg of protein/sample was digested at 25 °C for 12 h with lysyl endopeptidase (LysC, Wako Chemicals) at a 1:25; protein:protease ratio. Following LysC digestion the peptides in 2 M guanidine were diluted to 0.5 M guanidine with 100 mM EPPS, pH 8.0. The lysC peptides were digested with trypsin at 37 °C for 8 h (Promega) at a 1:50; protein:protease ratio. Then 1/10 volume of 45 mM dithiothreitol (DTT) was added to the cleared cell lysates and the reduction reaction was performed at 60 °C for 20 min on the second set of samples. The reaction was cooled down on ice for 10–15 min before the alkylation reaction. Iodoacetamide (IAA) based alkylation reaction was performed with 1/10 volume of 110 mM IAA and incubated at room temperature in the dark for 15 min. The protein in the 9 M urea buffer was diluted to 2 M with 20 mM HEPES, pH 8.0. Approximately 500 µg of protein/sample was digested at 37 °C for a minimum of 16 h (~ overnight) with trypsin (Promega, V5113) at a 1:50; protein: protease ratio. Sep-Pak Vac 1 cc (50 mg) tC18 cartridges (WAT054960) were used to clean-up the digested samples. The trypsin digested lysates were acidified with 1/20 volume of 20% trifluoroacetic acid (TFA) and centrifuged at 16,000$g$ for 15 min prior to the application on Sep-Pak columns. Sep-Pak columns were pre-wet with 1 ml of 100% acetonitrile and washed twice with 2 ml of 0.1% TFA solution. The clarified peptide solutions were loaded onto the washed columns. The columns were washed once with 3 ml of 0.1% TFA before elution of the peptides with 1 ml of the elution buffer twice (40% acetonitrile in 0.1% TFA elution buffer). Peptide amount per sample was quantified with peptide assay (Pierce, 23275) before samples were dried overnight using speed-vac. The resuspended peptides from set-1 were volume adjusted by adding 1 M EPPS, pH 8.0 in a 3:1 ratio (peptide volume:1 M EPPS volume; 250 mM EPPS final). 50 µg of peptide from each sample was labelled with 200 µg of TMTpro reagent resuspended in 5 µl, 100% acetonitrile. The peptides were incubated with TMTpro reagent for 3 h at 25 °C. TMTpro-labelled peptides were quenched with hydroxylamine (0.5% final) and acidified with trifluoroacetic acid (2% final). The samples were combined, desalted with 50 mg tC18 Sep-Paks, dried by vacuum.

TMT labelling was performed on 100 µg of peptides per sample for set-2. The dried peptides were resuspended in 100 µl 200 mM HEPES, pH 8.0 buffer. The TMT reagent was solubilized in 20 µl anhydrous acetonitrile and mixed with the peptide solution. Samples were pooled together once the TMT labeling efficiency reached > 98% in the label check experiment. The pooled sample was desalted with Sep-Pak C18 column and dried in the speed-vac.

The dried peptides from set-1 were resuspended in 0.1% TFA. Approximately 120 µg of peptide mix was subjected to orthogonal basic pH reverse phase fractionation on a 3x150 mm column packed with 1.9 µm Poroshell C18 material (Agilent, Santa Clara, CA) equilibrated with buffer A (5% acetonitrile in 10 mM ammonium bicarbonate, pH 8). Peptides were fractionated utilizing a 45 min linear gradient from 12% to 45% buffer B (90% acetonitrile in 10 mM ammonium bicarbonate, pH 8) at a flow rate of 0.8 ml/minute. 96 fractions were consolidated into 24 samples and vacuum dried. The samples were resuspended in 0.1% TFA desalted on StageTips and vacuum dried. Peptides were reconstituted in 5% formic acid + 5% acetonitrile for LC-MS3 analysis. The dried labelled peptide from set-2 mixture was fractionated by basic pH reversed-phase liquid chromatography on Agilent into 96 fractions and concatenated into 24 fractions. The experimental method used to fractionate the peptides was adopted from the previously published work[30]. The concatenated fractions were cleaned up with C18 spin tips (Pierce, 84850) and dried in speed-vac prior to LC–MS analysis.

## LC–MS and data analysis

The dried peptides were reconstituted in LC buffer A (2% acetonitrile/ 0.1% formic acid) on Dionex Ultimate 3000 RSLCnano system (Thermo Fisher) and Orbitrap Eclipse Tribrid MS (Thermo Fisher). The peptide separation was performed on 25 cm length and 75 µM diameter AURORA series column packed with 1.6 µM C18 material with pore size of 120 Å (Ion Opticks, IO2575011997). A linear LC gradient of 90 min with 2% to 30% buffer B (98% acetonitrile/0.1% formic acid) in buffer A (2% acetonitrile/0.1% formic acid) at flow rate of 400 nl min$^{-1}$. The sample analysis was performed using a multinotch MS3-TMT method and data dependent mode. The scan sequence started with FTMS1 spectra (resolution = 120,000; mass range ($m/z$) = 350–1,350; maximum injection time = 50 ms; normalized AGC target (%) = 250; dynamic exclusion = 35 s with a ±10 ppm mass tolerance window). In data dependent scans, the time between master scans was set up as 1 s with 2–6 as charge state filter. The selected peaks were fragmented via collision-induced dissociation (CID) in the ion trap (CID collision energy (%) = 35; maximum injection time = 100 ms; isolation window = 0.5 Da; normalized AGC target (%) = 150). Following ITMS2 acquisition, a real-time search (RTS) was performed to score peptides and select only the high scoring peptides for trigger synchronous-precursor-selection (SPS) MS3 quantitation. We also used a linear discriminant approach to assist the RTS step. For RTS, we used carbamidomethyl on cysteines (57.0215 Da) and TMTpro16plex (304.2071 Da) on lysines as static modifications. In variable modifications, oxidation on methionine (15.9949 Da) and TMT-pro16plex on tyrosines (304.2071 Da) were selected. The additional RTS parameters were: max missed cleavages = 1; max variable mods/ peptide = 1; enable FDR filtering = True; precursor neutral loss ($m/z$) = 0.0; enable protein close-out = True; max peptides per protein = 3; max search time = 35 ms. Up to 8 SPS precursors were further isolated and fragmented with high energy collision-induced dissociation (HCD) with Orbitrap analysis (HCD collision energy (%) = 40; resolution = 50,000; max injection time = 350 ms; normalized AGC target (%) = 250; isolation window = 1.2 Da).

The data were searched using comet[31] against a target decoy database that included Uniprot *Homo sapiens* protein sequences, contaminant and reversed sequences of proteins (version 2017/08). The search parameters were: peptide mass tolerance = 25 ppm; enzyme specificity = fully digested; allowed missed cleavages = 2; variable modifications = oxidation of methionine (15.9949 Da), TMTpro (tyrosine) = 304.2071 Da; fragment ion tolerance = 0.4; static modifications = carbamidomethyl on cysteines (57.0215 Da) and TMTpro16plex (304.2071 Da) on N-terminal and lysine residues. Peptide and protein level data was passed through 2% FDR separately following the previously published algorithm[32,33]. The searched dataset was further processed for TMT reporter ion intensity-based quantitation using Mojave algorithm[34] with an isolation width of 0.7.

## Peptide quantification and statistical analysis

PSMs were filtered out if they were from decoy proteins; from peptides with length less than 5; with isolation specificity less than 50%; with reporter ion intensity less than $2^8$ noise estimate; from peptides shared by more than one protein; with summed reporter ion intensity (across all sixteen channels) lower than 30,000. In the case of redundant

PSMs (that is, multiple PSMs in one MS run corresponding to the same peptide ion), only the single PSM with the least missing values or highest isolation specificity or highest maximal reporter ion intensity was retained for subsequent analysis. Quantification and statistical analysis were performed by MSstatsTMT v2.2.7, an open-source R/Bioconductor package[35]. Multiple fractions from the same TMT mixture were combined in MSstatsTMT. In particular, if the same peptide ion was identified in multiple fractions, only the single fraction with the maximal summation of reporter ion intensity was kept. MSstatsTMT generated a normalized quantification report across all the samples at the protein level from the processed PSM report. Global median normalization equalized the median of the reporter ion intensities across all the channels and TMT mixtures, to reduce the systematic bias between channels. The normalized reporter ion intensities of all the peptide ions mapped to a protein were summarized into a single protein level intensity in each channel and TMT mixture. For set-1, additional local normalization on the summarized protein intensities was performed to reduce the systematic bias between two TMT mixtures. For local normalization, we created an artefact reference channel by averaging over all the channels for each protein and TMT mixture. Then MSstatsTMT equalized the protein intensities in the reference channel of two TMT mixtures to the mean of the reference channels between the TMT mixtures. Then, it applied the corresponding shifts to the protein intensities in the remaining channels of each TMT mixture. MSstatsTMT performed differential abundance analysis for the normalized protein intensities. MSstatsTMT estimated $\log_2$(fold change) and the standard error by linear mixed effect model for each protein. The inference procedure was adjusted by applying an empirical Bayes shrinkage. To test the two-sided null hypothesis of no changes in abundance, the model-based test statistics were compared with the Student t-test distribution with the degrees of freedom appropriate for each protein and each dataset. The resulting $P$ values were adjusted to control the FDR with the method by Benjamini–Hochberg.

### Identification and characterization of novel cell-surface ligases

Given that RNF43 and ZNRF3 localize to the plasma membrane and have exposed ECDs, their domain structure was evaluated revealing the presence of an N-terminal signal peptide that has been associated with membrane integration as well as a transmembrane domain. As such, a list of known E3 ubiquitin ligases was evaluated using UniProtKB, the SignalP software and the DeepLoc software to identify putative cell-surface E3 ubiquitin ligases based on having a signal peptide, transmembrane domain and whether they have been shown or are predicted to localize to the plasma membrane. Among the identified proteins, we evaluated the following 11 ubiquitin ligases: RNF13, RNF43, RNF128, RNF130, RNF133, RNF148, RNF149, RNF150, RNF167, ZNRF3 and ZNRF4. Doxycycline-inducible pBind plasmids with IRES-eGFP encoding each of the 11 ligases with an N-terminal gD tag and C-terminal Flag tag were constructed and used to generate stable HT29, SW48 and ASPC1 cell lines. Cell-surface presentation of each ligase was evaluated using flow cytometry. In brief, parental and doxycycline-inducible gD–ligase-Flag HT29 cells were plated at 250,000 cells per well in 96-well round bottom plates. After 24 h cells were treated with 1 µg ml$^{-1}$ doxycycline for an additional 24 h to induce ligase expression. Cells were then prepared for FACS analysis by detaching in 5 mM EDTA, blocking in FACS buffer and stained using the Ultra-LEAF purified human IgG1 isotype control recombinant antibody or an anti-gD primary antibody at 10 µg ml$^{-1}$ for 1 h on ice followed by a Goat anti-human 647 secondary antibody incubation for 1 h on ice in the dark. Samples were then stained with LIVE/DEAD Fixable Violet Dead Cell Stain Kit, for 405 nm excitation at 1:1,000 for 30 min on ice and fixed using the Image-iT Fixative Solution (4% formaldehyde, methanol-free) (Thermo Fisher, R37814) for 15 min at room temperature. Plates were stored at 4 °C until further flow cytometry analysis. To evaluate doxycycline induction the IRES-eGFP signal was examined. The 647 MFI and percentage of gD + 647 positive cells were quantified from three independent experiments. As assay controls, parental cells were also evaluated. Additionally, unstained samples were processed in parallel and used for the gating strategy.

In parallel to flow cytometry analysis, HT29 and SW48 doxycycline-inducible gD–ligase-Flag cells were used to evaluate the degradative potential of each putative cell-surface ligase. Following 24 h doxycycline induction, cells were treated with 1–2 µg ml$^{-1}$ gD*IGF1R, gD*HER2 or gD*PD-L1 bispecific PROTABs or the respective NIST controls for 24–48 h. Cells were then lysed and prepared for western blot analysis to examine the total levels of each target upon bispecific antibody treatment.

### Statistics and reproducibility

Data are represented as mean ± s.e.m. unless otherwise stated. All experiments were reproduced at least twice unless otherwise indicated. None of the experiments were blinded and no statistical methods were used to pre-determine sample size for in vivo experiments. The number of animals used for each experiment is specified in the relevant figures and legends, and was estimated based on the variability in tumour take rate and growth observed for each model. With regards to randomization, for subcutaneous transplantation, mice were distributed among treatment groups when tumours reached a mean volume of approximately 200–400 mm$^3$. Prism versions 8.0 and 9.0 (GraphPad Software) were used.

### Schematic representations

All cartoons and schematics were generated with BioRender.com with the exception of the HiBiT-LgBiT Nano-luciferase technology and mouse schematics that were generated by Adobe Illustrator specifically for this manuscript.

### Reporting summary

Further information on research design is available in the Nature Research Reporting Summary linked to this paper.

## Data availability

Quantitative mass spectrometry proteomics data (raw data, metadata for experimental design, quantification results, and testing result) have been deposited to MassIVE (https://massive.ucsd.edu/ProteoSAFe/static/massive.jsp) with the dataset identifier MSV000089542. Login credentials: username, MSV000089542_reviewer; password, znrf3. RNA-seq data have been deposited at the Gene Expression Omnibus under accession GSE208372. Source data are provided with this paper.

## Code availability

All commercial and custom codes used to generate data presented in this study are available at https://doi.org/10.5281/zenodo.6855630.

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

**Acknowledgements** We thank D. Mandikian for performing antibody radiolabelling, the Genentech APAT group for high-throughput production and assembly of bispecific proteins, and C. M. Rose for his suggestions with setting up comet search parameters.

**Author contributions** F.d.S.e.M. and N.J.A conceived the study with input from F.J.d.S. and J.L. H.M., K.R., W.-T.K.T., Y.-S.K., J.H. and J.L. conceived, performed and analysed most of the experiments with assistance from C.W., P.Z., Y.T.K., K.D. and D.A. L.C.-A., E.J., A.D. and E.M. performed receptor copy number analysis. S.H. designed and engineered RNF43 and ZNRF3 expression constructs. I.L. designed all gD-tagged cell-surface ligase constructs. T.S. designed DmrA and DmrC constructs. D.S., H.X. and J.W. performed antibody discovery. I.K. performed small-scale bispecific antibody assembly and purification. M.G. and T.N. oversaw all bioinformatic analysis. N.K., M.H., D.K., A.G.M. and C.C. performed in vivo studies. B.B. performed in situ hybridization. P.S.-L. extracted protein lysates from all ex vivo samples. M.C., P.D. and S.P. supervised, performed and analysed all mass spectrometry-related data. The manuscript was written by H.M., W.-T.K.T., N.J.A. and F.d.S.e.M., and included contributions from all authors.

**Competing interests** All authors are current or former employees of Genentech.

**Additional information**
**Correspondence and requests for materials** should be addressed to Nicholas J. Agard or Felipe de Sousa e Melo.

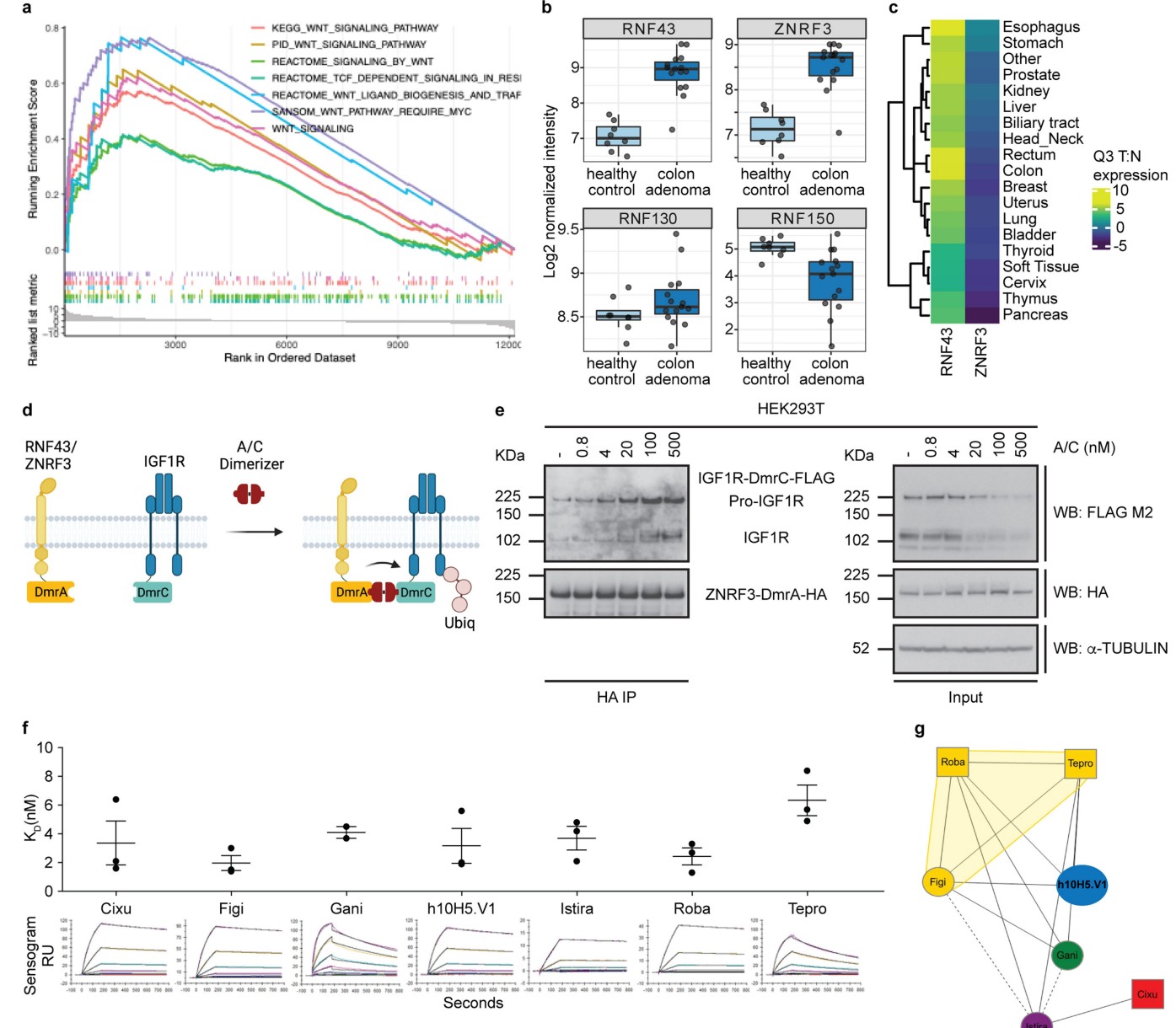

**Extended Data Fig. 1 | Dimerization of IGF1R to the Wnt-responsive E3 ubiquitin ligases RNF43 and ZNRF3 leads to IGF1R degradation.**
a) Comparative gene set enrichment analysis of wild type (WT) and APC truncation mutant (*Apc*^−/−^) colon organoids highlighting MSigDB C2 Wnt pathway gene sets from three independent experiments. b) RNA expression profile of indicated E3 ubiquitin ligases with predicted transmembrane domains in healthy control and colon adenomas. Box plots represent data distribution (minimum, first quartile, median, third quartile and maximum) with individual gene probes overlayed from 8 independent healthy control and 15 colon adenoma samples. c) Heatmap depicting RNF43 and ZNRF3 TCGA RNA expression profile in human tumors compared to their respective normal tissues. d) Schematic representation of iDimerize technology utilized to chemically induce the interaction of C-terminally DmrA-HA-tagged RNF43 or ZNRF3 with C-terminally DmrC-FLAG-tagged IGF1R following treatment with the A/C heterodimerizer. e) Western blot lysate analysis from HEK293T cells co-expressing C-terminally DmrA-HA-tagged ZNRF3 and C-terminally

DmrC-FLAG-tagged IGF1R subjected to HA immunoprecipitation (IP) following no treatment (-) or 24 h treatment with indicated A/C heterodimerizer concentrations. Precipitated and total DmrA-HA-tagged ZNRF3 and DmrC-FLAG-tagged IGF1R were detected. α-TUBULIN was used as a loading control. Data are representative of three independent experiments. f) Summary of Surface Plasmon Resonance (SPR) analysis of IGF1R bivalent antibodies binding to purified human IGF1R. Graph depicts bivalent antibody affinity to human IGF1R detected using Biacore T200 at 37 °C in HBSEP buffer with representative sensorgrams from three independent experiments. Data are presented as mean ± s.e.m. with values from biological repeats overlaid. For Gani one outlier data point was excluded. g) Epitope binning analysis of IGF1R bivalent antibodies binding to purified human IGF1R. For (f and g) IGF1R bivalent antibodies were tested (Cixu = Cixutumumab; Figi = Figitumumab; Gani = Ganitumab; h10H5.V1; Istira = Istiratumab; Roba = Robatumumab; Tepro = Teprotumumab). For gel source data, see Supplementary Fig. 2.

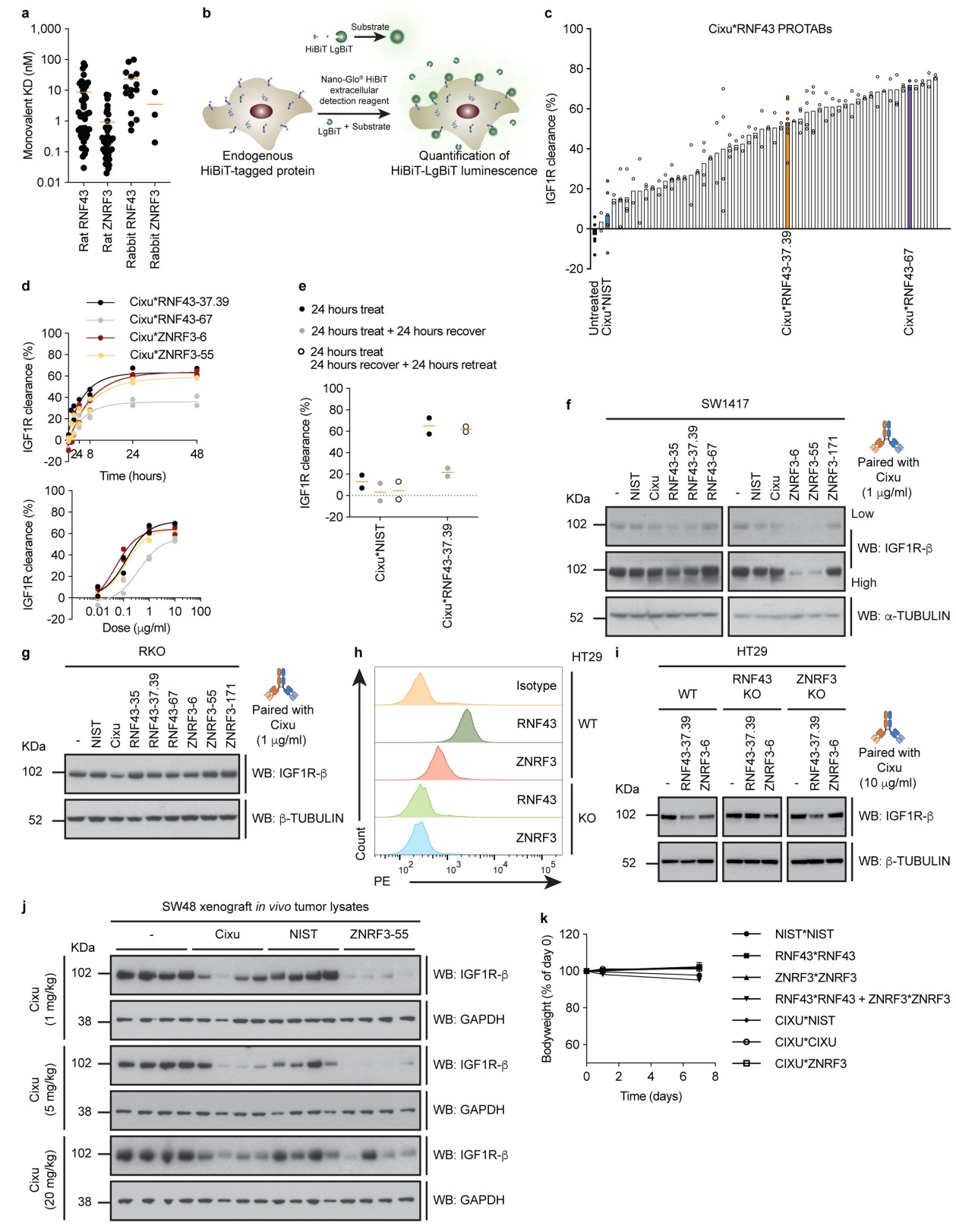

**Extended Data Fig. 2** | See next page for caption.

**Extended Data Fig. 2 | RNF43 and ZNRF3 bivalent antibody campaigns facilitate rational design of ligase-based bispecific PROTABs that tether endogenous RNF43 or ZNRF3 to IGF1R.** a) SPR analysis of RNF43 or ZNRF3 bivalent antibodies from rabbit and rat antibody campaigns binding to purified human RNF43 or ZNRF3 extracellular domain. Graph depicts $K_D$ values of individual campaign antibodies from one high throughput screen and line as the mean value per campaign. b) Schematic representation of HiBiT-LgBiT Nano-luciferase technology. Endogenous IGF1R was N-terminally tagged with HiBiT. Incubation of live cells with LgBiT and substrate allows HiBiT-LgBiT Nano-luciferase reconstitution and substrate catalysis. This can be quantified using luminescence as a proxy for IGF1R cell-surface levels. c) IGF1R cell-surface clearance in HT29 HiBiT-IGF1R knock-in (KI) cells subjected to LgBiT + substrate incubation following 24 h treatment with RNF43*IGF1R bispecific PROTABs. Graph depicts IGF1R clearance (%) from one screening campaign with values from technical repeats overlaid. Assay controls and PROTABs used subsequently are highlighted. d, e) IGF1R cell-surface clearance in HT29 HiBiT-IGF1R KI cells subjected to LgBiT + substrate incubation following treatment with indicated antibodies. Graphs depict IGF1R clearance (%) time kinetics (*top*) or dose response (*bottom*) (d) and impact of dosing, recovery and redosing (e) from two independent experiments. Data are presented as non-linear curves (d and e) and mean (e) with values from biological repeats overlaid. f, g) Western blot lysate analysis from SW1417 (f) or RKO (g) cells left untreated (-) or subjected to indicated antibodies for 48 h. Endogenous IGF1R-β was detected. α and β-TUBULIN were used as loading controls. Data are representative of three independent experiments. h) Flow cytometry histograms of PE signal in HT29 wildtype (WT), RNF43 Knock Out (KO) or ZNRF3 KO cells stained with an antibody against RNF43, ZNRF3 or the matched isotype control from two independent experiments. i) Western blot lysate analysis from HT29 WT, RNF43 KO or ZNRF3 KO cells that were left untreated (-) or subjected to indicated antibodies for 24 h. Endogenous IGF1R-β protein was detected. β-TUBULIN was used as a loading control. Data are representative of two independent experiments. j) Western blot lysate analysis from SW48 xenograft in vivo tumours derived from mice left untreated (-) or subjected to indicated antibodies for 72 h. Endogenous IGF1R-β was detected. GAPDH was used as a loading control. N = 4 animals per group. k) Mice bodyweight following treatment with indicated antibodies for 7 days. Graph depicts changes of bodyweight (% of day 0) over time from four animals per group. Data are presented as mean ± s.e.m. For gel source data, see Supplementary Fig. 2.

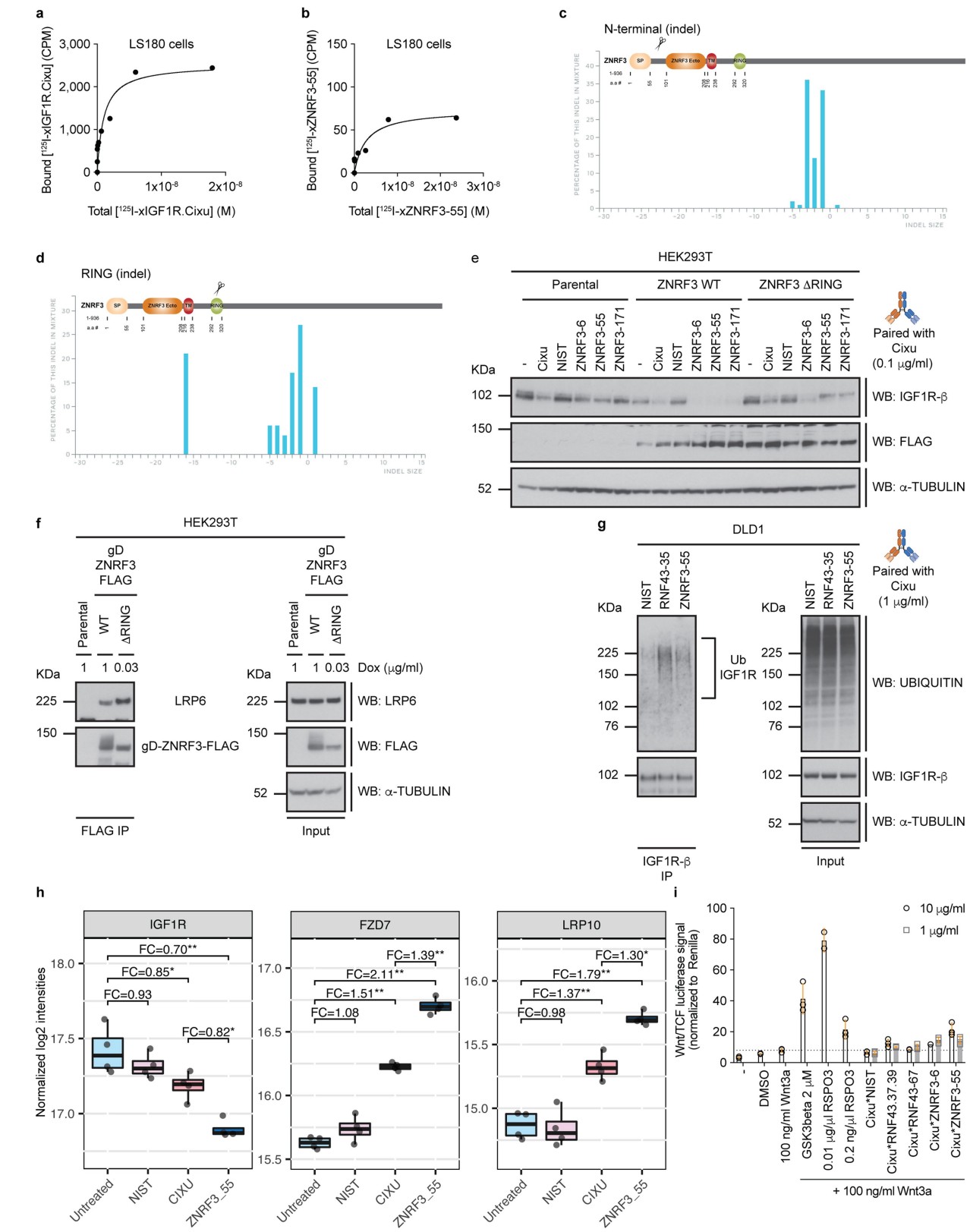

**Extended Data Fig. 3** | See next page for caption.

**Extended Data Fig. 3 | ZNRF3 cell-surface levels and catalytic activity mediate IGF1R ubiquitylation and degradation induced by ZNRF3\*IGF1R bispecific PROTABs.** a, b) Anti-IGF1R Cixu (a) or anti-ZNRF3 ZNRF3–55 (b) bivalent antibodies saturation assay in LS180 cells. Graphs depict bound target antibody in counts per minute (CPM) versus concentration of total antibody. Data are presented as one-site specific binding curves fitted to mean values from three technical repeats. c, d) Indel representation in genetically engineered HT29 ZNRF3 N-term(i) (c) or HT29 ZNRF3 RING(i) (d) cells. Graphs depict the percentage of various indel sizes in HT29 pooled cells from two independent analyses. e) Western blot lysate analysis from doxycycline-treated parental HEK293T, doxycycline-inducible WT or delta RING (ΔRING) gD-ZNRF3-FLAG cells left untreated (-) or subjected to indicated antibodies for 24 h. Endogenous IGF1R-β and exogenous gD-ZNRF3-FLAG were detected. Data are representative of two independent experiments. f) Western blot lysate analysis from parental HEK293T, doxycycline-inducible WT or ΔRING gD-ZNRF3-FLAG cells subjected to FLAG immunoprecipitation (IP). Precipitated and total gD-ZNRF3-FLAG and LRP6 were detected. Data are representative of three independent experiments. g) Western blot lysate analysis from DLD1 cells subjected to denaturing IGF1R-β IP following treatment with indicated antibodies for 2 h. Precipitated and total endogenous IGF1R-β and ubiquitin were detected. Data are representative of two independent experiments. For (e-g) α-TUBULIN was used as a loading control. h) Normalized protein intensities for depicted proteins across various antibody treatments in SW48 cells. Box plots represent data distribution (minimum, first quartile, median, third quartile and maximum) with biological repeats overlaid from four independent experiments. The lower whisker is the smallest normalized log2 intensities or equal to lower bound of box - 1.5*IQR (Interquantile range = 75% quantile – 25% quantile). The upper whisker is the largest normalized log2 intensities or equal to upper bound of box +1.5*IQR (minimum, first quartile, median, third quartile and maximum). Fold changes are the difference between the means of four biological repeats per condition; * = adjusted p-value < 0.05, ** = adjusted p-value < 0.005. To test the two-sided null hypothesis of no changes in abundance, the model-based test statistics were compared with the Student t-test distribution with the degrees of freedom appropriate for each protein and each dataset by MSstatsTMT R package. The resulting p-values were adjusted to control the FDR with the method by Benjamini-Hochberg. IGF1R, adj.pvalue = 0.00004 for ZNRF3_55 vs Untreated, adj.pvalue = 0.032 for CIXU vs Untreated, adj.pvalue = 0.6256 for NIST vs Untreated, adj. pvalue = 0.0329 for ZNRF3_55 vs CIXU, LRP10, adj.pvalue = 5.27e-7 for ZNRF3_55 vs Untreated, adj.pvalue = 0.0007 for CIXU vs Untreated, adj. pvalue = 0.9309 for NIST vs Untreated, adj.pvalue = 0.0093 for ZNRF3_55 vs CIXU. FZD7, adj.pvalue = 3.76e-10 for ZNRF3_55 vs Untreated, adj. pvalue = 1.46e-6 for CIXU vs Untreated, adj.pvalue = 0.3828 for NIST vs Untreated, adj.pvalue = 2.34e-5 for ZNRF3_55 vs CIXU. i) Wnt signaling activity measured in TCF luciferase HEK293 cells subjected to indicated antibodies. RSPO3 and GSK3beta were used as positive Wnt agonists. Data are presented as mean ± s.d. with values from individual technical repeats from one experiment overlaid. Graph is representative of two independent experiments. For gel source data, see Supplementary Fig. 2.

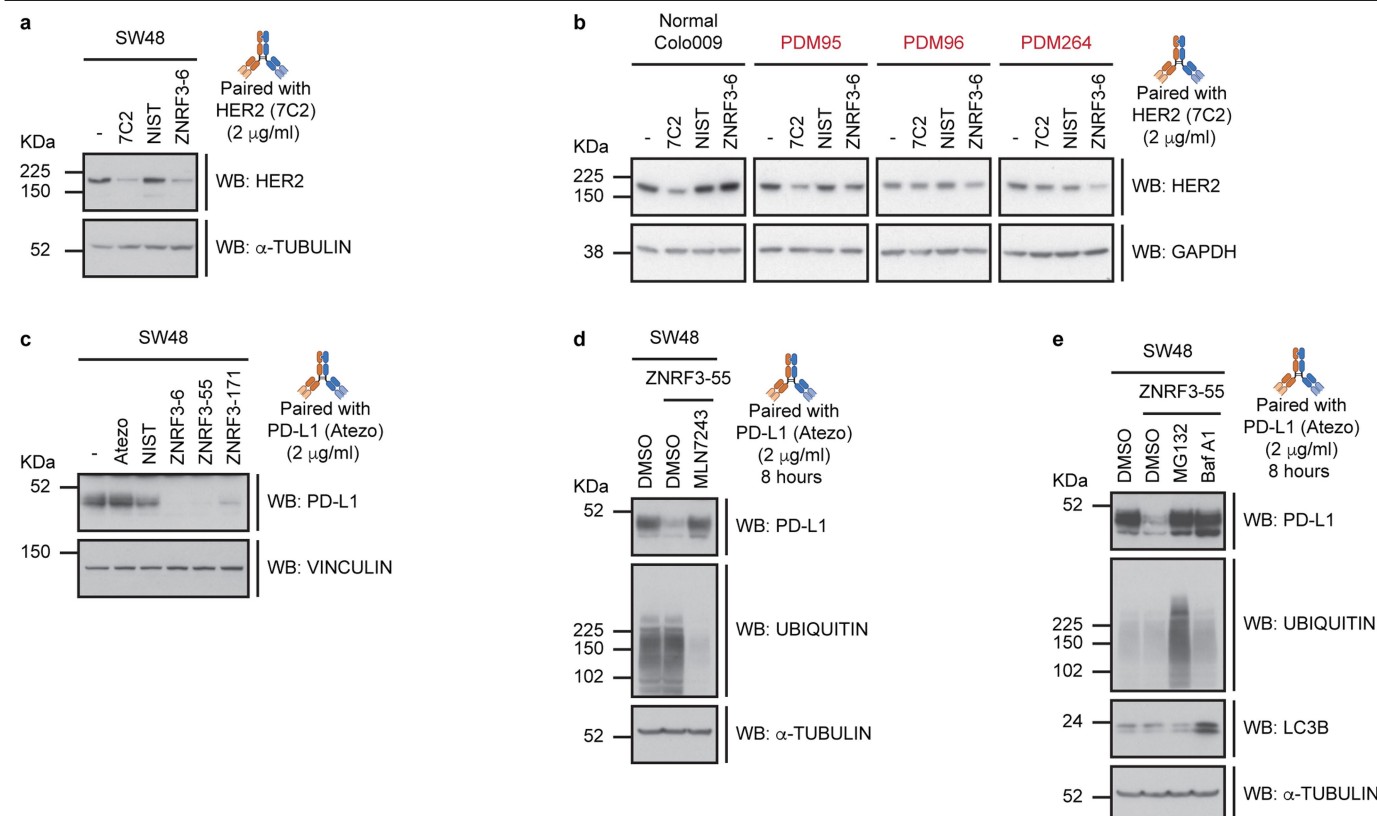

**Extended Data Fig. 4 | ZNRF3 PROTABs targeting endogenous HER2 and PD-L1 induce target degradation.** a) Western blot lysate analysis from SW48 cells left untreated (-) or subjected to indicated antibodies for 48 h. Data are representative of three independent experiments. b) Western blot lysate analysis of healthy or tumor-derived organoids treated with indicated antibodies for 24 h. Data are representative of two independent experiments. For (a and b) endogenous HER2 was detected. α-TUBULIN and GAPDH were used as loading controls. c) Western blot lysate analysis from SW48 cells left untreated (-) or subjected to indicated antibodies for 24 h. Endogenous PD-L1 was detected. VINCULIN was used as a loading control. Data are representative of three independent experiments. d) Western blot lysate analysis from SW48 cells subjected to DMSO or E1 inhibitor MLN7243 2 h pre-treatment followed by the ZNRF3–55*PD-L1 bispecific PROTAB for 6 h. Endogenous PD-L1 and ubiquitin were detected. e) Western blot lysate analysis from SW48 cells subjected to DMSO, the proteasome inhibitor MG132 or the lysosomal pathway inhibitor Bafilomycin A1 (Baf A1) 2 h pre-treatment followed by the ZNRF3–55*PD-L1 bispecific PROTAB for 6 h. Endogenous PD-L1, ubiquitin and LC3B were detected. For (d and e) α-TUBULIN was used as a loading control. Data are representative of two independent experiments. For gel source data, see Supplementary Fig. 2.

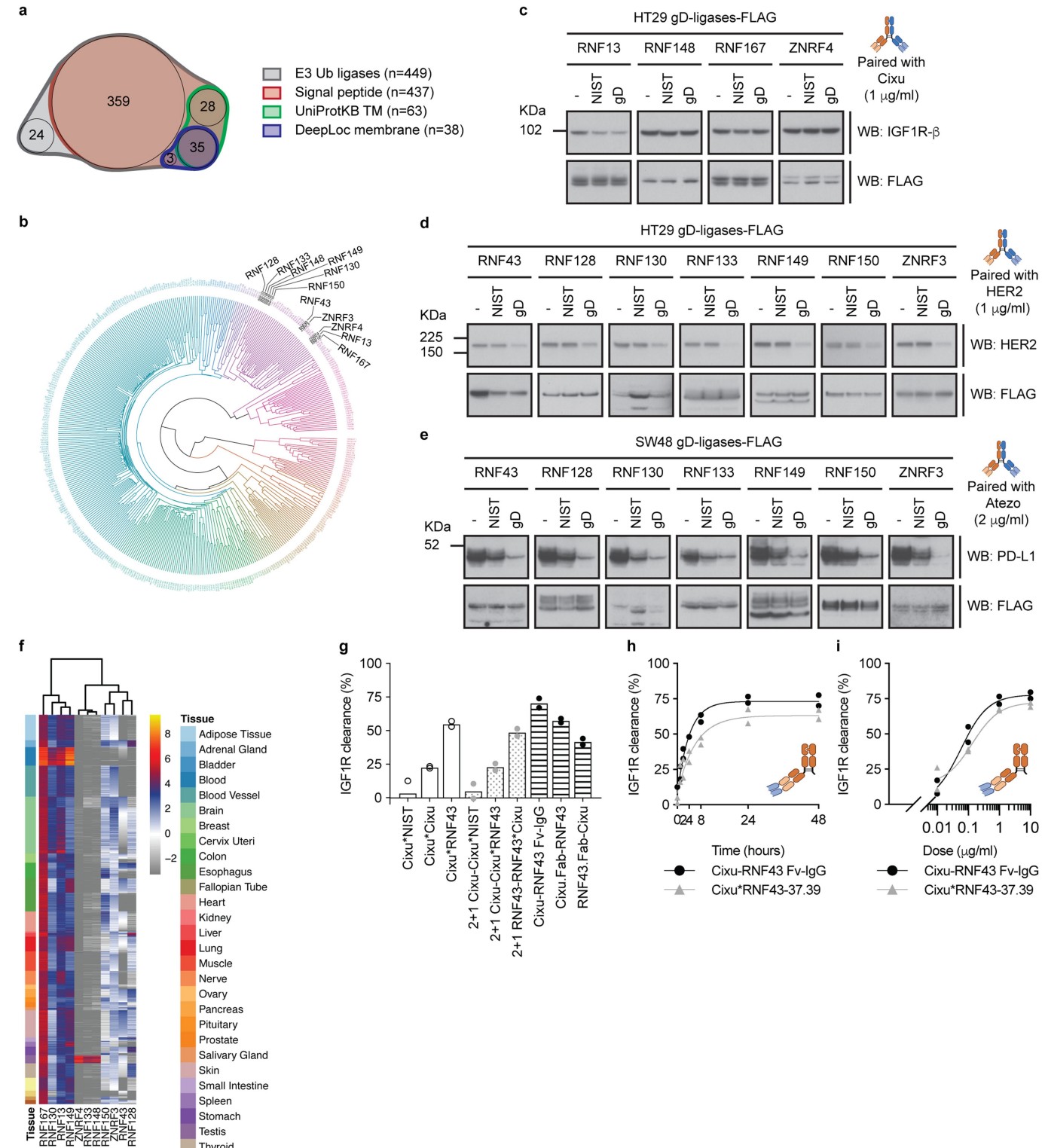

**Extended Data Fig. 5 |** See next page for caption.

**Extended Data Fig. 5 | Identification of cell-surface E3 ubiquitin ligases and altering PROTABs format facilitate platform expansion.** a) Venn diagram depicting bioinformatics analysis to identify putative cell-surface E3 ubiquitin ligases based on having a signal peptide, transmembrane domain and whether they have been shown or are predicted to localize to the plasma membrane. b) Dendrogram of known E3 ubiquitin ligases clustered based on sequence homology highlighting a subset of putative cell-surface E3 ubiquitin ligases that were tested for cell-surface expression and applicability as PROTAB degraders. c) Western blot lysate analysis from indicated doxycycline-treated HT29 doxycycline-inducible gD-ligase-FLAG cells following no treatment (-) or 24 h incubation with indicated antibodies. Endogenous IGF1R-β and exogenous gD-ligase-FLAG proteins were detected. Data are representative of three independent experiments. d) Western blot lysate analysis from indicated doxycycline-treated HT29 doxycycline-inducible gD-ligase-FLAG cells following no treatment (-) or 24 h incubation with indicated antibodies. Endogenous HER2 and exogenous gD-ligase-FLAG proteins were detected. Data are representative of two independent experiments. e) Western blot lysate analysis from indicated doxycycline-treated SW48 doxycycline-inducible gD-ligase-FLAG cells following no treatment (-) or 48 h incubation with indicated antibodies. Endogenous PD-L1 and exogenous gD-ligase-FLAG proteins were detected. Data are representative of three independent experiments. f) Heatmap depicting expression of indicated cell-surface E3 ubiquitin ligases across normal tissues (source GTEX). Data is z-score normalized. g) IGF1R cell-surface clearance in HT29 HiBiT-IGF1R KI cells subjected to LgBit + substrate incubation following treatment with indicated antibodies for 24 h. Graph depicts IGF1R clearance (%) from two independent experiments. Data are presented as mean with values from biological repeats overlaid. h, i) IGF1R cell-surface clearance in HT29 HiBiT-IGF1R KI cells subjected to LgBiT + substrate incubation following treatment with indicated RNF43*IGF1R PROTAB bispecific or one-arm Fv-IgG format. Graphs depict IGF1R clearance (%) time kinetics (h) or dose response (i) from two independent experiments. Data are presented as non-linear curves with values from biological repeats overlaid. For gel source data and molecular weight sizes of gD-ligase-FLAG in (c–e), see Supplementary Fig. 2.

# Reporting Summary

## Statistics

For all statistical analyses, confirm that the following items are present in the figure legend, table legend, main text, or Methods section.

| n/a | Confirmed | |
|---|---|---|
| ☐ | ☒ | The exact sample size (*n*) for each experimental group/condition, given as a discrete number and unit of measurement |
| ☐ | ☒ | A statement on whether measurements were taken from distinct samples or whether the same sample was measured repeatedly |
| ☐ | ☒ | The statistical test(s) used AND whether they are one- or two-sided *Only common tests should be described solely by name; describe more complex techniques in the Methods section.* |
| ☒ | ☐ | A description of all covariates tested |
| ☒ | ☐ | A description of any assumptions or corrections, such as tests of normality and adjustment for multiple comparisons |
| ☐ | ☒ | A full description of the statistical parameters including central tendency (e.g. means) or other basic estimates (e.g. regression coefficient) AND variation (e.g. standard deviation) or associated estimates of uncertainty (e.g. confidence intervals) |
| ☐ | ☒ | For null hypothesis testing, the test statistic (e.g. *F*, *t*, *r*) with confidence intervals, effect sizes, degrees of freedom and *P* value noted *Give P values as exact values whenever suitable.* |
| ☒ | ☐ | For Bayesian analysis, information on the choice of priors and Markov chain Monte Carlo settings |
| ☒ | ☐ | For hierarchical and complex designs, identification of the appropriate level for tests and full reporting of outcomes |
| ☒ | ☐ | Estimates of effect sizes (e.g. Cohen's *d*, Pearson's *r*), indicating how they were calculated |

*Our web collection on statistics for biologists contains articles on many of the points above.*

## Software and code

Policy information about availability of computer code

| Data collection | RNA-sequencing (RNA-seq) libraries were sequenced on Illumina HiSeq 2500 sequencers<br>Flow cytometry was performed on a BD FACSymphony A3 Cell Analyzer and using the BD Diva acquisition software (version 9.0)<br>Immunohistochemistry pictures were acquired with an NIKON A1R using N3 elements software. |
|---|---|
| Data analysis | FlowJo 10.7.1 for flow cytometry analysis<br>Prismv8 and Prismv9<br>R (v4.1.0)<br>hg38 and mm9 gene models from Genentech's RefSeq-derived internal database IGIS (v4.0)<br>seqinr (v4.2)<br>Biostrings (v2.60.2)<br>hgu133plus2.db (v3.13.0)<br>annotationTools (v1.66.0)<br>Limma (v3.48.3)<br>SummarizedExperiment (v1.22.0)<br>IRanges (v2.26.0)<br>org.Hs.eg.db (v3.13.0)<br>SignalP (v5.0)<br>DeepLoc (v1.0)<br>nVennR (v0.2.3)<br>ggplot2 (v3.3.5) |

plotly (v4.9.4.1)
msigdbr (v7.4.1) enrichplot (v1.12.2) clusterProfiler (v4.0.5) pheatmap
(v1.0.12) protr (v1.6-2) circlize (v0.4.13) dendextend (v1.15.1)

All commercial and custom codes used to generate data presented in
this study are available DOI: 10.5281/zenodo.6855630

For manuscripts utilizing custom algorithms or software that are central to the research but not yet described in published literature, software must be made available to editors and reviewers. We strongly encourage code deposition in a community repository (e.g. GitHub). See the Nature Portfolio guidelines for submitting code & software for further information.

## Data

Policy information about availability of data

All manuscripts must include a data availability statement. This statement should provide the following information, where applicable:
- Accession codes, unique identifiers, or web links for publicly available datasets
- A description of any restrictions on data availability
- For clinical datasets or third party data, please ensure that the statement adheres to our policy

Epithelial Systems Biology Laboratory (ESBL) Human E3 Ubiquitin Ligases (accessed 6/2/2021); https://esbl.nhlbi.nih.gov/Databases/KSBP2/Targets/Lists/E3-ligases/ Genentech Release v6.0.2 of the GTEx project data (http://www.gtexportal.org/). This data currently contains 9120 RNA-Seq samples from 549 different donors. Quantitative mass spectrometry proteomics data (raw data, metadata for experimental design, quantification results, and testing result) have been deposited to MassIVE (https://massive.ucsd.edu/ProteoSAFe/static/massive.jsp) with the data set identifier MSV000089542. Login credentials: Username: MSV000089542_reviewer. Password: znrf3
RNA seq data is available under GSE208372.
All source data is provided or accessible upon request.

## Human research participants

Policy information about studies involving human research participants and Sex and Gender in Research.

| Reporting on sex and gender | Use the terms sex (biological attribute) and gender (shaped by social and cultural circumstances) carefully in order to avoid confusing both terms. Indicate if findings apply to only one sex or gender; describe whether sex and gender were considered in study design whether sex and/or gender was determined based on self-reporting or assigned and methods used. Provide in the source data disaggregated sex and gender data where this information has been collected, and consent has been obtained for sharing of individual-level data; provide overall numbers in this Reporting Summary.  Please state if this information has not been collected. Report sex- and gender-based analyses where performed, justify reasons for lack of sex- and gender-based analysis. |
|---|---|
| Population characteristics | Describe the covariate-relevant population characteristics of the human research participants (e.g. age, genotypic information, past and current diagnosis and treatment categories). If you filled out the behavioural & social sciences study design questions and have nothing to add here, write "See above." |
| Recruitment | Describe how participants were recruited. Outline any potential self-selection bias or other biases that may be present and how these are likely to impact results. |
| Ethics oversight | Identify the organization(s) that approved the study protocol. |

Note that full information on the approval of the study protocol must also be provided in the manuscript.

## Field-specific reporting

Please select the one below that is the best fit for your research. If you are not sure, read the appropriate sections before making your selection.

☒ Life sciences   ☐ Behavioural & social sciences   ☐ Ecological, evolutionary & environmental sciences

For a reference copy of the document with all sections, see nature.com/documents/nr-reporting-summary-flat.pdf

## Life sciences study design

All studies must disclose on these points even when the disclosure is negative.

| Sample size | Statistical methods were not used to predetermine sample size. Sample size followed common standard of n=2 or more biological replicates. The number of animals used for each experiments was estimated based on the variability in tumor take and tumor growth observed for each models in previous in house studies. Number of animals are described in relevant figure legends or relevant figures and were all equal or above n=4 per group. |
|---|---|
| Data exclusions | No data was excluded from the manuscript. For Extended Data Fig. 1f outlier value excluded from graph is provided in corresponding source file. |

| | |
|---|---|
| Replication | All experiments presented are representative of at least two biological repeats that showed reproducible results unless otherwise indicated. Exceptions include the high throughput binding analysis of RNF43 and ZNRF3 bivalent antibodies generated through the rabbit and rat antibody campaigns presented in Fig. 2 and Extended Data Fig. 2. Due to the large number of campaign antibodies, these were produced at small scale, which precluded biological repeats. For these, all antibody binding was orthogonally validated through flow cytometry with two independent biological experiments. Copy number analysis presented in Extended Fig. 3a and 3b represent technical replicates of one biological experiment. The low copy number of these receptors is at the lower limit of detection that can only be detected with high efficiency in 125-I labeling of the antibody, which varies between assays. To minimize batch effect, the entire experiment across all cell lines and receptors was performed in single batch. |
| Randomization | For animal studies animals were randomized after tumors had reached a size of ~ 400 mm3 to receive a single intraperitoneal injection of the PROTABs. No other experiments were randomized. To counter for potential batch effect, all in vitro treatment were performed on the same plate. |
| Blinding | Blinding during experimental procedures were not required because experimental conditions could be easily identified from the data itself. |

# Reporting for specific materials, systems and methods

We require information from authors about some types of materials, experimental systems and methods used in many studies. Here, indicate whether each material, system or method listed is relevant to your study. If you are not sure if a list item applies to your research, read the appropriate section before selecting a response.

## Materials & experimental systems

| n/a | Involved in the study |
|---|---|
| ☐ | ☒ Antibodies |
| ☐ | ☒ Eukaryotic cell lines |
| ☒ | ☐ Palaeontology and archaeology |
| ☐ | ☒ Animals and other organisms |
| ☒ | ☐ Clinical data |
| ☒ | ☐ Dual use research of concern |

## Methods

| n/a | Involved in the study |
|---|---|
| ☒ | ☐ ChIP-seq |
| ☐ | ☒ Flow cytometry |
| ☒ | ☐ MRI-based neuroimaging |

## Antibodies

| | |
|---|---|
| Antibodies used | Akt (pan) (40D4) Mouse mAb; Cell Signaling; 2920S; Lot 8; Dilution 1:1,000<br>Anti-HA High Affinity (3F10); Roche Diagnostics; 11867423001; Lot 45715900; Dilution 1:1,000<br>Anti-mouse IgG, HRP-linked Antibody; Cell Signaling; 7076S; Lot 36; Dilution 1:5,000<br>Anti-rabbit IgG, HRP-linked Antibody; Cell Signaling; 7074S; Lot 30; Dilution 1:5,000<br>Anti-rat IgG, HRP-linked Antibody; Cell Signaling; 7077S; Lot 14; Dilution 1:5,000<br>GAPDH (14C10) Rabbit mAb (HRP Conjugate); Cell Signaling; 3683; Lot 4; Dilution 1:1,000<br>Goat anti-Human IgG (H+L) Secondary Antibody, Alexa Fluor 647; Invitrogen; A21445; Lot 2339821; Dilution 1:1,000<br>Goat anti-Mouse IgG (H+L) Secondary Antibody, HRP; Invitrogen; 32430; Lot VD301382; Dilution 1:5000<br>Goat anti-Rabbit IgG (H+L) Secondary Antibody, HRP; Invitrogen; 32460; Lot VE30198; Dilution 1:5000<br>HER2/ErbB2 (D8F12) XP® Rabbit mAb; Cell Signaling; 4290S; Lot 6; Dilution 1:1,000<br>IGF-I Receptor β (D23H3) XP®Rabbit mAb; Cell Signaling; 9750S; Lot 5-7; Dilution:1,000<br>IGF1R (1H7), APC; eBioscience; 17-8849-42; Lot 2330467; Dilution 1:20<br>IRDye® 800CW Donkey anti-Mouse IgG Secondary Antibody; LI-COR; 926-32212; Lot D10414-15 and D00930-09; Dilution 1,5000<br>IRDye® 800CW Goat anti-Rabbit IgG Secondary Antibody; LI-COR; 926-32211; Lot D10629-12 and D01110-10; Dilution 1,5000<br>LC3B antibody; Novus Biologicals; NB100-2220; Lot EU/EU-3; Dilution 1:1,000<br>LIVE/DEAD™ Fixable Violet Dead Cell Stain Kit, for 405 nm excitation; Thermo Fisher; L34964; Lot 2208471; Dilution 1,000<br>LRP6 (C5C7) Rabbit mAb; Cell Signaling; 2560S; Lot 11; Dilution 1:1,000<br>Monoclonal ANTI-FLAG® M2 antibody produced in mouse; Sigma-Aldrich; F3165; Lot SLCG2330 and SLBT6752; Dilution 1:1,000<br>Monoclonal Anti-α-Tubulin antibody produced in mouse; Sigma-Aldrich; T9026; Lot 099M4773V; Dilution 1:10,000<br>PD-L1 (E1L3N®) XP® Rabbit mAb; Cell Signaling; 13684S; Lot 18; Dilution 1:1,000<br>Phospho-Akt (Ser473) Antibody; Cell Signaling; 9271S; Lot 15; Dilution 1:1,000<br>Phospho-IGF-I Receptor β (Tyr1135/1136)/Insulin Receptor β (Tyr1150/1151) (19H7) Rabbit mAb; Cell Signaling; 3024S; Lot 11; Dilution 1:1,000<br>Phospho-S6 Ribosomal Protein (Ser235/236) Antibody; Cell Signaling; 2211S; Lot 23; Dilution 1:1,000<br>Polyclonal Rabbit Anti-Human c-erbB-2 Oncoprotein; Agilent (DAKO); A0485; Lot 20083958; Dilution 1:1,000<br>S6 Ribosomal Protein (5G10) Rabbit mAb; Cell Signaling; 2217S; Lot 10; Dilution 1:1,000<br>Ubiquitin monoclonal antibody (P4G7-H11); Enzo; ADI-SPA-203-F; Lot 04062139; Dilution 1:1,000<br>Ultra-LEAF™ Purified Human IgG1 Isotype Control Recombinant Antibody; BioLegend; 403502; Lot B322065; Dilution 10 ug/ml<br>Vinculin Antibody; Cell Signaling; 4650S; Lot 5; Dilution 1:1,000<br>β-Tubulin (9F3) Rabbit mAb; Cell Signaling; 2128L; Lot 11; Dilution 1:1,000 |
| Validation | All commercial antibodies utilized in this study were selected based on provided manufacturer's validation data. For in house antibodies, such as RNF43, ZNRF3 and anti-gD used for bispecific antibody generation, WB and flow cytometry antibodies were validated in house by ELISA, SPR, dot blots (data not shown) as well as WB (data not shown) and flow cytometry (Extended Data Fig. 2 |

and Fig. 3 for representative RNF43 and ZNRF3 in house bivalent antibodies; Fig. 4 for anti-gD antibody) using cell lines overexpressing the proteins of interest and in the case of RNF43 and ZNRF3 KO cells were also used for flow cytometry validation.

# Eukaryotic cell lines

Policy information about cell lines and Sex and Gender in Research

| | |
|---|---|
| Cell line source(s) | HEK293T, LS1034, KM12, DLD1, HT115, LS180, LS513, SW1417, HT55, GP2D, RKO, COLO678, SW48 and ASPC1 lines and primary human organoids were obtained from ATCC and maintained by Cell Central, an in house cell line repository, at Genentech, Inc. Primary murine organoid cultures were also used (described in methods under "organoids" subsection). |
| Authentication | All lines are authenticated using standard genotyping methods. |
| Mycoplasma contamination | All cell lines acquired from Cell Central are routinely tested for mycoplasma and no mycoplasma contamination was reported for parental cell lines utilized. Genetically engineered lines were generated using parental lines obtained from cell central and were not further tested for mycoplasma contamination. |
| Commonly misidentified lines (See ICLAC register) | No misidentified lines were used in this study. |

# Animals and other research organisms

Policy information about studies involving animals; ARRIVE guidelines recommended for reporting animal research, and Sex and Gender in Research

| | |
|---|---|
| Laboratory animals | WT B57BL/6 mice (000664) and NOD.Cg-Prkdcscidll2rgtm1Wjl/SzJ (NSG) (colony 005557) mice were purchased from the Jackson Laboratory. Sprague Dawley rats were obtained from Charles River, Hollister, CA. White rabbits were obtained from WORC (Western Oregon Rabbit Co). Females of 6 to 12 weeks old were used for experiments. Standard housing conditions, including dark/light cycle, ambient temperature and humidity were used. |
| Wild animals | The study did not involve wild animals. |
| Reporting on sex | *Indicate if findings apply to only one sex; describe whether sex was considered in study design, methods used for assigning sex. Provide data disaggregated for sex where this information has been collected in the source data as appropriate; provide overall numbers in this Reporting Summary. Please state if this information has not been collected. Report sex-based analyses where performed, justify reasons for lack of sex-based analysis.* |
| Field-collected samples | The study did not involve samples collected from the field. |
| Ethics oversight | Animal studies were approved by Genentech's Institutional Animal Care and Use Committee and adhere to the NRC Guidelines for the Care and Use of Laboratory Animals. |

Note that full information on the approval of the study protocol must also be provided in the manuscript.

# Flow Cytometry

## Plots

Confirm that:

☒ The axis labels state the marker and fluorochrome used (e.g. CD4-FITC).

☒ The axis scales are clearly visible. Include numbers along axes only for bottom left plot of group (a 'group' is an analysis of identical markers).

☒ All plots are contour plots with outliers or pseudocolor plots.

☒ A numerical value for number of cells or percentage (with statistics) is provided.

## Methodology

| | |
|---|---|
| Sample preparation | 96-well plates cell surface staining detailed protocol: <br> • Day1: <br> o Plate cells in 96-well round bottom plate at 250,000 cells/well (adjust depending on cell line used) <br> • Day2: <br> o Perform media exchange (+ dox at 1 ug/ml if using inducible cell line) <br> • Day3: <br> o After 48 hours of cell plating and 24 hours of dox induction (if applicable) prepare cells for FACS staining <br> o Aspirate media using gel loading tips attached to 8-well aspirator (to avoid cell suction and loss) <br> o Wash cells in 100 µl/well PBS <br> o Aspirate PBS using gel loading tips attached to 8-well aspirator (to avoid cell suction and loss) <br> o Detach cells using 5 mM EDTA (this gives a better signal for cell surface staining compared to trypsin and accutase) <br> o Incubate for 15 minutes at 37 °C (adjust depending on cell line used) |

o Examine cells under microscope to ensure that they are fully detached and pipet cells up and down to facilitate the process
o Add 100 μl/well media to detached cells, mix by pipetting and spin at 1,200 rpm for 5 minutes at 4°C
o Aspirate media using gel loading tips attached to 8-well aspirator (to avoid cell suction and loss)
o Resuspend cell pellets in 100 μl/well FACS buffer and block by incubating for 10 minutes on ice
o After incubation, spin at 1,200 rpm for 5 minutes at 4°C
o Aspirate media using gel loading tips attached to 8-well aspirator (to avoid cell suction and loss)
o Resuspend cell pellets in 100 μl/well 1ry antibody solution (prepared in FACS buffer) and incubate for 1 hour on ice
o After incubation, spin at 1,200 rpm for 5 minutes at 4°C
o Aspirate media using gel loading tips attached to 8-well aspirator (to avoid cell suction and loss)
o Resuspend cell pellets in 100 μl/well FACS buffer and spin at 1,200 rpm for 5 minutes at 4°C
o Repeat wash step a total of 3 times
o Resuspend cell pellets in 100 μl/well 2ry antibody solution (prepared in FACS buffer) and incubate for 1 hour on ice covered in foil
o After incubation, spin at 1,200 rpm for 5 minutes at 4°C
o Aspirate media using gel loading tips attached to 8-well aspirator (to avoid cell suction and loss)
o Resuspend cell pellets in 100 μl/well FACS buffer and spin at 1,200 rpm for 5 minutes at 4°C
o Repeat wash step a total of 2 times
o Resuspend cell pellets in 100 μl/well LIVE/DEAD fixable dead cell stain solution (prepared in FACS buffer or PBS) and incubate for 30 minutes on ice covered in foil
o After incubation, spin at 1,200 rpm for 5 minutes at 4°C
o Aspirate media using gel loading tips attached to 8-well aspirator (to avoid cell suction and loss)
o Resuspend cell pellets in 100 μl/well FACS buffer and spin at 1,200 rpm for 5 minutes at 4°C
o Resuspend cell pellets in 100 μl/well fixative solution and incubate for 15 minutes at RT covered in foil
o After incubation, spin at 1,200 rpm for 5 minutes at 4°C
o Aspirate media using gel loading tips attached to 8-well aspirator (to avoid cell suction and loss)
o Resuspend cell pellets in 100 μl/well FACS buffer and spin at 1,200 rpm for 5 minutes at 4°C
o Repeat wash step a total of 2 times
o Resuspend cell pellets in 100 μl/well FACS buffer
o Analyze cells immediately using flow cytometry or store covered in foil at 4°C for later processing
Primary antibodies
Anti-Human IGF-1R-APC, Clone 1H7, Invitrogen/eBioscience, Catalog # 17-8849-42, Lot 2172983
Ultra-LEAF purified human IgG1 isotype control; Biolegend; 403502
hRNF43.RNF43-129HC.hIgG1; In house; Req ID: 547062; PUR ID: 612194
hZNRF3.ZNRF3-275HC.hIgG1; In house; Req ID: 547073; PUR ID: 612064

Secondary antibody
Goat anti-Human (H+L) cross-absorbed 2ry antibody, Alexa Fluor 647; Invitrogen; A21445
Goat anti-human IgG (H+L) 2ry antibody, PE; Invitrogen; PA1-86078

Live/dead dye
LIVE/DEAD fixable violet dead cell stain, for 405 nm excitation; Thermo Fisher Scientific; L34964A

Fixative solution
Image-iT fixative solution (4 % formaldehyde, methanol free);; Thermo Fisher Scientific; FB002

| | |
|---|---|
| Instrument | BD FACSCelestaTM Flow Cytometer #660344 |
| Software | BD FACSDivaTM for collection and FlowJo 10.7.1 for analysis. |
| Cell population abundance | Flow cytometry was mainly used to analyze cell-surface staining of various proteins |
| Gating strategy | For endogenous IGF1R cell-surface staining presented throughout the manuscript and exogenous ligase cell-surface staining presented in Fig.4 cell gating was performed as follows: FSC-A/SSC-A manual scatter gate on cell population followed by FSC-H/FSC-W and SSC-H/SSC-W manual gate on single cells. Live cells were selected in applicable experiments. Detailed gating strategies used are outlined in Supplementary Figure 3. |

☒ Tick this box to confirm that a figure exemplifying the gating strategy is provided in the Supplementary Information.

