## [Peer Review file. · Nature]

Manuscript Title: Antibody targeting of E3 ubiquitin ligases for receptor degradation

Reviewer Comments & Author Rebuttals

Reviewer Reports on the Initial Version:

Referees' comments:

Referee #2 (Remarks to the Author):

The authors demonstrate that dimeric antibodies (PROTABs) can induce the dimerization of cell surface E3 ligases with cell surface receptors, resulting in ubiquitination and degradation of the receptor, using IGFR1 as an example. Interestingly they demonstrate that RNF43 and ZNRF3 are upregulated in colorectal cancer, thus suggesting that tumour specific degradation may be achievable. The authors first demonstrate that chemically induced dimerization between tagged IGFR1 and RNF43 induced loss of IGFR1 before using antibodies to exogenously expressed RNF43/ZNRF3 to demonstrate degradation of IGFR1.

The authors then went on to generate antibodies to RNF43/ZNRF3 and show that they could induce IGFR1 degradation using endogenous ligases, across multiple cell lines and that degradation of IGFR1 is advantageous over inhibition using Cixu antibody, in vitro and in vivo. Mutations in the ligase inhibit degradation going some way to demonstrate that the degradation is UPS dependent, although this is confounded by the fact that the cixu antibody itself also appears to induce the degradation of IGFR1 (as many inhibitory antibodies do). The authors demonstrate that PROTABs induced ubiquitination of the target and that degradation can be somewhat rescued by inhibition of the proteasome or the lysosome.

Importantly the authors go on to demonstrate that other cell surface E3 ligases can induce the degradation of IGFR1. HER2 and PD-L1, using an endogenous epitope antibody approach. Finally the authors demonstrate various methods of antibody engineering can improve activity of PROTABs.

This is an interesting paper, which is generally well controlled and describes a platform technology with significant potential as a platform for therapeutic antibody development. One major concern for publication in Nature is the novelty, since (as the authors note) the Wells lab recently reported on RNF43 recruiting bivalent antibodies for PD-L1 degradation. Despite this, this article may be suitable for publication if the below major concerns are addressed.

Firstly, I do not think the authors have demonstrated the generality of this approach suitably to make their claims regarding multiple ligase or targets since the majority of the data is focused on IGFR1 and the only data for other ligases relies on epitope tags. For publication in Nature, I would like to see the authors demonstrate that antibodies can be raised against an additional cell surface ligase and that it can be applied to degrade another cell surface protein – ideally one which has not been degraded (HER2 and PD-L1 degradation have both previously been reported) by other approaches previously (eg. a GPCR).

Additionally, the claims regarding tissue/tumor selectivity are not supported and despite lots of discussion, no tissue/tumor specificity is demonstrated. This is a long-standing problem in the field of targeted protein degradation and I urge the authors to do the experiments to address this question since it appears this could be a feasible approach.

The mechanism of action for PROTABs is unclear, how does the protein get removed from the cell surface? Is it truly a dual lysosome/proteasome degradation process? Additional experiments to determine the molecular mechanism are required.

The lack of differentiation between Cixu mediated degradation and PROTAB induced degradation is a concern throughout the manuscript – in places it's challenging to draw conclusions when the negative control also induces degradation.

Finally, no selectivity data is presented to demonstrate that only the targeted proteins are degraded by the PROTABs, quantitative proteomics data should be included to demonstrate selectivity of degradation.

Other minor concerns:

The PD-L1 WB in Fig. 4e are not of high enough quality to draw any conclusions (particularly for some of the ligases).

There appears to be poor correlation between the WB data presented in Fig. 1h/l and the data presented in Fig 1g. This should be commented on and explained.

No mention of TRIM-Away, another antibody mediated protein degradation approach.

Unclear labelling: Fig. 1 h/l are labelled WT but it's not WT protein

The heat map in Fig 2.g is too small and hard to interpret

Referee #3 (Remarks to the Author):

In the current manuscript, Marei and colleagues describe an interesting method to mediate the degradation of transmembrane receptors (mainly receptor tyrosine kinases) by repurposing transmembrane E3 ubiquitin ligases to specific targets with a bispecific antibody that bridges the E3 ligase to their novel target. The designed PROTABs approach differs from previously known PROTACs and LYTACs because it targets proteins from outside the cell, circumventing issues concerning permeability and pharmacokinetics that complicate the latter approaches. In doing so, they re-direct two WNT target E3 ubiquitin ligases known to be upregulated in CRC (because of WNT pathway aberrations) to mediate the lysosomal degradation of IGF1R and also of other targets like HER2 and PD-L1.

In general, the results presented are convincing and support the conclusions of the authors regarding the efficacy of the technique in mediating the degradation of the desired target.

The structure function analysis of the RNF43/ZNRF3 does not add much to what is already known about these surface E3 ligases. The study does not go beyond a proof of concept, i.e. the observation

that retargeting these two E3 ligases to other surface proteins than their natural targets (the Frizzleds), decreases surface expression of these non-natural targets. A series of other E3 ligases is also tested, but only very superficially analyzed.

Major comments:

-Given that multiple other strategies to retarget ubiquitin ligases have been described, one would like to see firm proof that this approach is superior. What is direly missing from the current paper is in vivo evidence that the approach actually works: i.e. that targeting surface molecules on CRC cells actually results in tumor clearance. Two models immediately spring to mind: 1) Targeting EGFR family members in well-chosen human PDX models would be an obvious experimental CRC platform. 2) Targeting PD-L1 in an immunocompetent mouse model, using isogenic tumor organoids (as described previously by Batlle et al). Now, all analyses are performed for a maximum of 72h, mostly biochemical in nature and are almost exclusively in vitro.

-Non-transformed cells at the base of crypts closely resemble APC-mutant cells in expressing high levels of ZNRF3/RNF43 as well as the dependence on tyrosine kinase receptor signaling. One would like to see if there exists a therapeutic window between crypts and neoplasms.

-Retargeting RNF43 and ZNRF3 away from Frizzleds is predicted to strongly hyperactivate physiological Wnt signals (this actually mimicks the mode of action of R-spondins). It is highly likely that this will occur in the (healthy) intestines of treated individuals (mouse, man). One definitely does not want to hyperactivate the Wnt pathway in individuals with CRC. I would have expected a detailed assessment of this potential complication in vivo, comparing the effects on normal crypts vs. adeno(carcino)mas

Other remarks

-It is difficult to follow the experiments jump between cell lines and models. For instance, it is clear that the methods are developed in the HEK293T cell line (easier to manipulate), and consequently validated in CRC cell lines like DLD1, HT115 and SW1417 among others. However, it is rather confusing to present the results of the validation in a different line depending on the figure and the experiment (for example Fig. 2f, 2h, 3c and 3e all represent different lines).

-Why did the author select IGF1R as target of their repurposed transmembrane E3 ubiquitin ligases? In the context of CRC, the impact of IGF1R on CRC biology is rather unclear. Other transmembrane proteins (such as EGFR, HER2, PD-L1 and RNF43 C-term truncated mutants) are much more obvious candidates. IGF1R is more commonly implied in tumors of the liver (which have high frequencies of CTNNB1 mutations). Perhaps that would be a good model system to evaluate the clinical potential of the PROTABs targeting IGF1R.

-Why did the authors selected CRC-derived immortalized cell line as model system to evaluate the efficacy of their technique in vitro? Despite being easy to manipulate and culture, their reliability as relevant pre-clinical model remains questionable. In Fig.1 the authors present a pilot experiment performed on mouse organoids engineered to express multiple mutations found during CRC

progression. Having access to such model systems can be fundamental to evaluate the efficacy of the technique in more relevant settings (CRC patients-derived organoids).

-Why was the antibody Cixutumumab selected for the targeting of the ECD of IGF1R and not Roba? According to the Western Blot in Fig. 1h and 1i, Roba displays a better clearance of IGF1R from the membrane.

-Occasional degradation of the IGF1R is observed for the bivalent Cixutumumab formation, which is enhanced in the DLD1, KM12 and the HT115 lines. Is this effect comparable if using any of the other IGF1R targeting antibodies?

-Figure 2i, was the injection of SW48 subcutaneous or orthotopic? Was any adverse effect observed during treatment with the PROTABs? Did the authors evaluate the inhibition of downstream effectors (like pAKT) as well after the in vivo delivery of the PROTABs?

-Blocking either the proteasomal or lysosomal protein degradation systems results in partial rescue of the PROTABs-induced IGF1R degradation (Fig. 3g). To the authors knowledge, does IGF1R possess a recycling mechanism to the cell surface for the non-degraded protein? Such mechanism is known for other membrane receptors like EGFR which recycles back to the membrane after escaping lysosomal-mediated degradation.

-Fig. 4c, why the amount of IGF1R detected in the negative controls is significantly different in different PROTABs (for example RNF43 vs RNF149)? Also, why a double band is detected in RNF150 of the same figure?

-Fig.4e, the degradation of PD-L1 is not clear from the figure. Can the authors present a more convincing picture?

-In general, all figures including Western Blots data should be presented with the original pictures as source data in the supplementary material attached to the main manuscript (For example, in Fig. 4e there is a band that has been cut from the picture).

Author Rebuttals to Initial Comments:

Response to reviewers' comments for Marei *et al.* "Repurposing E3 ubiquitin ligases as cell surface protein degraders using Proteolysis Targeting Antibodies" Nature ID manuscript # 2021-11-17506A

Referee #2 (Remarks to the Author):

The authors demonstrate that dimeric antibodies (PROTABs) can induce the dimerization of cell surface E3 ligases with cell surface receptors, resulting in ubiquitination and degradation of the receptor, using IGFR1 as an example. Interestingly they demonstrate that RNF43 and ZNRF3 are upregulated in colorectal cancer, thus suggesting that tumour specific degradation may be achievable. The authors first demonstrate that chemically induced dimerization between tagged IGFR1 and RNF43 induced loss of IGFR1 before using antibodies to exogenously expressed RNF43/ZNRF3 to demonstrate degradation of IGFR1.

The authors then went on to generate antibodies to RNF43/ZNRF3 and show that they could induce IGFR1 degradation using endogenous ligases, across multiple cell lines and that degradation of IGFR1 is advantageous over inhibition using Cixu antibody, in vitro and in vivo. Mutations in the ligase inhibit degradation going some way to demonstrate that the degradation is UPS dependent, although this is confounded by the fact that the cixu antibody itself also appears to induce the degradation of IGFR1 (as many inhibitory antibodies do). The authors demonstrate that PROTABs induced ubiquitination of the target and that degradation can be somewhat rescued by inhibition of the proteasome or the lysosome.

Importantly the authors go on to demonstrate that other cell surface E3 ligases can induce the degradation of IGFR1. HER2 and PD-L1, using an endogenous epitope antibody approach. Finally the authors demonstrate various methods of antibody engineering can improve activity of PROTABs.

This is an interesting paper, which is generally well controlled and describes a platform technology with significant potential as a platform for therapeutic antibody development. One major concern for publication in Nature is the novelty, since (as the authors note) the Wells lab recently reported on RNF43 recruiting bivalent antibodies for PD-L1 degradation. Despite this, this article may be suitable for publication if the below major concerns are addressed.

We thank the reviewer for their kind comments and enthusiasm on key impacts of our paper.

1) Firstly, I do not think the authors have demonstrated the generality of this approach suitably to make their claims regarding multiple ligase or targets since the majority of the data is focused on IGFR1 and the only data for other ligases relies on epitope tags. For publication in Nature, I would

like to see the authors demonstrate that antibodies can be raised against an additional cell surface ligase and that it can be applied to degrade another cell surface protein – ideally one which has not been degraded (HER2 and PD-L1 degradation have both previously been reported) by other approaches previously (eg. a GPCR).

We concur with this reviewer that including additional targets would bolster the claim of the generality of our platform. As such, we have now generated ZNRF3 PROTABs paired with various antibodies to target HER2, PD-L1 and have also included FZD5/8 (GPCR) in an effort to cover a broad range of receptor families. Consistent with our data on IGF1R/ZNRF3 PROTABs, degradation of various target receptors can be achieved in colon cancer cell lines, patient-derived primary CRC organoids and *in vivo*. There are two points we would like to further highlight:

1. In the case of PD-L1, we have observed deep and near complete degradation of the protein both *in vitro* and *in vivo* while minimal degradation was achieved using the bivalent or bispecific control (Figure 4b, Extended data figure 4c). This further demonstrates that, for particular targets, our platform can induce *de novo* receptor degradation that distinguishes itself from classical bivalent antibodies.
2. As there are no reliable anti-FZD antibodies for western blot analyses, we resorted to cell surface clearance detection by flow cytometry. Importantly, since most available flow cytometry detection antibodies recognize an overlapping epitope with the FZD arm used for our PROTAB (2919, dual FZD5/8 antibody) we have generated a FZD5 specific monoclonal antibody that binds to a different epitope than the PROTAB and enables accurate cell surface detection after PROTAB treatment. This data is now summarized in Figure 4e. Given the paucity of antibodies inhibiting GPCRs, we believe this highlights a novel approach to address a well validated class of drug-targets.

Lastly, we have also demonstrated the PROTAB technology can be extended to another target, EpCAM. Cells treated with EpCAM/Ligase PROTABs showed that RNF133 and RNF149 are able to induce modest EpCAM clearance (Rebuttal figure 1).

Rebuttal figure 1. EpCAM cell surface clearance in HT29 cells harboring indicated doxycycline inducible gD-ligase-FLAG expression constructs following 24 hours incubation with gD*EpCAM bispecific PROTAB. PROTABs at 1 µg/ml and 10 µg/ml. Graphs represent percentage of EpCAM cell surface clearance from two independent experiments.

2) Additionally, the claims regarding tissue/tumor selectivity are not supported and despite lots of discussion, no tissue/tumor specificity is demonstrated. This is a long-standing problem in the field of targeted protein degradation and I urge the authors to do the experiments to address this question since it appears this could be a feasible approach.

We thank the reviewer for this perspective. We agree that this point was not adequately addressed in the first version and as such we have attempted to demonstrate the selectivity we ascribed to the technology using two separate approaches:

- 1. Firstly, we evaluated *in vivo* degradation of various targets in tumor bearing mice and compared tumor receptor degradation to corresponding normal colon, where these ligases are typically expressed. Please note that the IGF1R PROTAB affinities for both antigens are preserved across species. Remarkably, as we now report in Figure 2i, Extended data figure 2j and Figure 4a, 4b, we observed potent degradation of various receptors in tumors, with minimal impact in normal colon (Figure 2i). This highlights that the elevated ligase expression in the pathological setting can indeed be leveraged to drive selective target degradation.**
- 2. Second, we have compared the activity of ZNRF3-based IGF1R and HER2 PROTABs in both normal and cancer-derived human colon organoids as these more faithfully recapitulate human physiology and disease compared to traditional cancer cell lines. In line with our data presented in numerous cancer cell lines, we demonstrate strong PROTAB activity in multiple CRC derived cancer organoids but critically, little to no activity in a normal colon organoid. In contrast, the bivalent antibodies targeting either HER2 or IGF1R showed indiscriminate degradation in both normal and cancer organoids (Figure 2j, Extended data figure 4b).**

Collectively, we believe these data illustrate the tumor selective nature of the ZNRF3 PROTABs in colon cancer compared to its normal tissue counterpart.

3) The mechanism of action for PROTABs is unclear, how does the protein get removed from the cell surface? Is it truly a dual lysosome/proteasome degradation process? Additional experiments to determine the molecular mechanism are required.

We agree with the reviewer that understanding the mechanism of action of this technology is important. Prior to delving into our efforts to address this comment, we would like to emphasize that degradation of receptors using PROTABs is relatively slow compared to cytosolic PROTACs, with efficient degradation occurring between 16-48 hours^{1,2}. This slow mechanism of action complicates the task of separating lysosomal and proteasomal mechanisms as extended proteasomal inhibition can drive lysosomal dysfunction, as revealed by elevated LC3B lipidation in the previous Figure 3g. Hence, it is possible that spillover

inhibition of the lysosome could contribute to the rescue of IGF1R degradation following PROTAB treatment. To probe this further, we selected a different cell line, SW48, in which substantial IGF1R degradation was observed as early as 6-8 hours post PROTAB treatment (Rebuttal figure 2). Indeed, shorter inhibition with MG132 results in less evident accumulation of lipidated LC3B, yet IGF1R rescue was still clearly observed with both proteasomal and lysosomal inhibition (Figure 3f replacing old Figure 3g).

To generalize this observation beyond IGF1R, we also evaluated the degradation process of an additional target, PD-L1. Similar to IGF1R, we found that both proteasomal and lysosomal inhibition with MG132 and BafA1, respectively, rescued PD-L1 degradation (Extended data figure 4e). Hence, we conclude that PROTAB activity likely relies on a dual lysosome/proteasome degradation process.

Rebuttal figure 2. Western blot analysis of lysates from SW48 that were treated with the indicated antibodies at the depicted time points. Endogenous IGF1R-β protein levels were detected. α-TUBULIN was used as a loading control. Data are representative of two independent experiments.

4) The lack of differentiation between Cixu mediated degradation and PROTAB induced degradation is a concern throughout the manuscript – in places it’s challenging to draw conclusions when the negative control also induces degradation.

Though RTKs are well validated antibody targets, with signaling pathways we anticipated might be altered by efficient degradation, as the reviewer highlighted, degradation mediated by bivalent antibodies is common. IGF1R is no exception and we found that many bivalent IGF1R antibodies lead to cell surface clearance or degradation in cells (Rebuttal figure 3). Indeed, the mechanism of action of Cixutumumab is in part dependent on degradation³. As such, we have tried to control for PROTAB activities in two ways: first, via using a monovalent binder to the target that does not recruit the ligase (NIST-bispecific antibodies) to assess the mechanistic impact of recruiting a ligase, and second, using bivalent antibodies to directly assess potential therapeutic superiority. As detailed below, we have expanded on our initial assessment by including antigens not readily degraded with bivalent antibodies, in a way that we believe will address the reviewer’s concern:

1. We demonstrated that PD-L1 can be effectively and specifically degraded *in vitro* and *in vivo* using a PROTAB compared to the corresponding bivalent antibody or PROTAB NIST control (Figure 4b, Extended data figure 4c) highlighting the distinctive activity of PROTAB versus bivalent antibody, at least for a subset of targets. Similar results are also seen for FZD suggesting that conversion of non-internalizing antibodies to degraders is a general finding.
2. In the context of targets that display degradation through bivalent blocking antibodies, such as HER2 and IGF1R, we now show that a key distinction between bivalent and PROTAB antibodies is in fact the tumor specific targeting of PROTABs as opposed to bivalent antibodies that do not discriminate between normal and tumor tissues. To this end and as described above, we have evaluated IGF1R degradation *in vivo* comparing tumor and normal tissues (Figure 2i). In addition, we show preferential PROTAB activity in cancer organoids compared to normal organoids while bivalent antibodies degrade in both contexts (IGF1R in Figure 2j, HER2 in Extended data figure 4b).

Finally, no selectivity data is presented to demonstrate that only the targeted proteins are degraded by the PROTABs, quantitative proteomics data should be included to demonstrate selectivity of degradation.

We thank the reviewer for raising this point. To address this comment, we have now performed global proteomics analysis to evaluate the specificity of our PROTABs. We have used quantitative mass spectrometry to perform a proteome-wide characterization of PROTAB degradation specificity in cells treated with IGF1R PROTABs or bivalent Cixutumumab at 24 hours post treatment. A significant reduction of IGF1R levels was induced by the ZNRF3 PROTAB, and to a lesser extent by the bivalent antibody (Extended data figure 3h). Changes in the expression levels of other proteins as a result of IGF1R degradation were also observed (Figure 3g, the full dataset has been deposited to MassIVE (<https://massive.ucsd.edu/ProteoSAFe/static/massive.jsp>) with the data set identifier MSV000089542. Login credentials: (Username: MSV000089542_reviewer. Password: znrf3). Because these co-degraded proteins were also detected in Cixutumumab treated cells 24 hours post treatment, we reasoned that they may be secondary events to IGF1R pathway inhibition rather than specifically targeted by the PROTAB. Interestingly, we also detected an increase in peptides that identified the endogenous substrates of ZNRF3, namely the Wnt receptor family members LRP and FZD. Hence, our proteomics data validates the specificity and mechanism of action of PROTAB as it concomitantly reveals the stabilization of the endogenous substrate of ZNRF3 with specific decrease of the protein of interest targeted by the PROTAB.

Other minor concerns:

The PD-L1 WB in Fig. 4e are not of high enough quality to draw any conclusions (particularly for some of the ligases).

To address the reviewer's comment, we have repeated this experiment in another cell line that expresses higher PD-L1 levels. We have thus replaced the original panel with the new data (Extended data figure 5e). Additionally, as highlighted above we now also validate PD-L1 degradation endogenously using ZNRF3-based PD-L1 PROTABs (Figure 4b and Extended data figure 4c)

There appears to be poor correlation between the WB data presented in Fig. 1h/I and the data presented in Fig 1g. This should be commented on and explained.

We thank the reviewer for pointing out this discrepancy. For consistency, we wanted to use HEK293T cells throughout Figure 1. Unfortunately, these cells have a relatively low expression of IGF1R compared to other cell lines tested, which was particularly problematic to detect with flow cytometry. To this end, we used HT29 cells for flow cytometry in Figure 1g and HEK293T for western blot validation in Figure 1 h and i. As we have described in Figure 2e, treatment with Cixu led to variable degradation across many cell lines. Hence, it is possible that differences between HEK293T and HT29 could explain the relatively modest correlation between flow and western blot data. We have added a comment in the manuscript to discuss this conundrum: *Of note, cell surface clearance did not always translate into degradation for some bispecific antibodies of similar affinity against IGF1R suggesting that the IGF1R epitope, and therefore the geometry of the ternary complex, may influence target degradation.*

No mention of TRIM-Away, another antibody mediated protein degradation approach.

We apologize for this omission and have now adjusted the text accordingly to describe TRIM-Away in the context of our PROTAB technology.

Unclear labeling: Fig. 1 h/I are labeled WT but it's not WT protein

This has been corrected

The heat map in Fig 2.g is too small and hard to interpret

To address the reviewer's comment, we have adapted this figure accordingly (Figure 2e).

Referee #3 (Remarks to the Author):

In the current manuscript, Marei and colleagues describe an interesting method to mediate the degradation of transmembrane receptors (mainly receptor tyrosine kinases) by repurposing

transmembrane E3 ubiquitin ligases to specific targets with a bispecific antibody that bridges the E3 ligase to their novel target. The designed PROTABs approach differs from previously known PROTACs and LYTACs because it targets proteins from outside the cell, circumventing issues concerning permeability and pharmacokinetics that complicate the latter approaches. In doing so, they re-direct two WNT target E3 ubiquitin ligases known to be upregulated in CRC (because of WNT pathway aberrations) to mediate the lysosomal degradation of IGF1R and also of other targets like HER2 and PD-L1.

In general, the results presented are convincing and support the conclusions of the authors regarding the efficacy of the technique in mediating the degradation of the desired target.

The structure function analysis of the RNF43/ZNRF3 does not add much to what is already known about these surface E3 ligases. The study does not go beyond a proof of concept, i.e. the observation that retargeting these two E3 ligases to other surface proteins than their natural targets (the Frizzleds), decreases surface expression of these non-natural targets. A series of other E3 ligases is also tested, but only very superficially analyzed.

We thank the reviewer for their kind comments and enthusiasm on key impacts of our paper.

Major comments:

-Given that multiple other strategies to retarget ubiquitin ligases have been described, one would like to see firm proof that this approach is superior. What is direly missing from the current paper is in vivo evidence that the approach actually works: i.e. that targeting surface molecules on CRC cells actually results in tumor clearance. Two models immediately spring to mind: 1) Targeting EGFR family members in well-chosen human PDX models would be an obvious experimental CRC platform. 2) Targeting PD-L1 in an immunocompetent mouse model, using isogenic tumor organoids (as described previously by Batlle et al). Now, all analyses are performed for a maximum of 72h, mostly biochemical in nature and are almost exclusively in vitro.

We agree with the reviewer that a comparative analysis of the various targeted protein degradation technologies is indeed appealing and of utmost importance. However, we feel that this goes beyond the scope of our manuscript, which revolves around the development and comprehensive characterization of a novel technology for protein degradation rather than establishing a therapeutic per se. While we believe that this approach can be widely applicable, identification of therapeutic pathways benefitting from targeted degradation is a substantial undertaking. Indeed, it has taken nearly twenty years for PROTACs to advance molecules through the clinic. Nevertheless, we believe that the additional characterization we have now performed in the revised manuscript will enable the technology to compare favorably to recent work published in Nature- e.g. LYTACs.

In particular, we have taken various points raised by the reviewer into consideration as our revised manuscript now contains:

- 1. Long term *in vitro* treatment to demonstrate superior efficacy of IGF1R PROTAB compared to the bivalent antibody (Figure 2g)**
- 2. Tumor specific targeting of IGF1R (Figure 2j) and HER2 (Extended Figure 4b) in cancer versus normal colon organoids**
- 3. Tumor degradation of various targets *in vivo* (Figure 2i, Figure 4a, Figure 4b)**
- 4. A comprehensive analysis of Wnt agonism *in vitro* and *in vivo* after PROTAB treatment (Figure 3h, Extended data figure 3i)**

-Non-transformed cells at the base of crypts closely resemble APC-mutant cells in expressing high levels of ZNRF3/RNF43 as well as the dependence on tyrosine kinase receptor signaling. One would like to see if there exists a therapeutic window between crypts and neoplasms.

We thank the reviewer for raising this important point, which relates to the therapeutic index of our platform and the tumor selectivity also raised by the other referee. We believe that this is a central point in our revised manuscript and an important distinctive aspect of the PROTAB platform compared to other degradation strategies. More specifically, we have addressed this comment using two separate approaches:

- 1. First, we evaluated *in vivo* degradation of various targets in tumor bearing mice and compared tumor receptor degradation to matching normal colon, where these ligases are typically expressed. Remarkably, and as we now report in Figure 2i, we could demonstrate potent degradation of IGF1R, yet with minimal impact in normal colon further highlighting the tumor specificity of our strategy, at least in the case of CRC and ZNRF3 PROTABs.**
- 2. Second, we have compared the activity of IGF1R (Figure 2j) and HER2 (Extended data figure 4b) PROTABs in both normal and cancer derived human colon organoids as these more faithfully recapitulate human physiology and disease compared to traditional cancer cell lines. In line with our data presented in numerous cancer cell lines, we demonstrate strong PROTAB activity in various CRC derived cancer organoids but critically, little to no activity in normal colon organoids. More importantly the bivalent antibodies targeting either HER2 or IGF1R showed indiscriminate degradation in both normal and cancer organoids. Collectively we believe these data illustrate the tumor selective nature of the ZNRF3 PROTABs in colon cancer compared to its normal tissue counterpart.**

-Retargeting RNF43 and ZNRF3 away from Frizzleds is predicted to strongly hyperactivate physiological Wnt signals (this actually mimicks the mode of action of R-spondins). It is highly

likely that this will occur in the (healthy) intestines of treated individuals (mouse, man). One definitely does not want to hyperactivate the Wnt pathway in individuals with CRC. I would have expected a detailed assessment of this potential complication *in vivo*, comparing the effects on normal crypts vs. adeno(carcino)mas

We thank the reviewer for raising this important point that warrants further discussion. Firstly, the PROTABs strategy presented here redirects either RNF43 *or* ZNRF3 and as such is unlikely to act as a full RSPO mimetic, which would interfere with both cell surface ligases simultaneously. In line with this, conditional knockout of RNF43 or ZNRF3 alone in the gut does not manifest any intestinal phenotype while only the double knockout results in crypt hyperplasia through hyperactivation of Wnt signaling ⁴. Secondly, recent exciting data in clonal competition in the gut has suggested that promoting Wnt signaling in this tissue, at least transiently, can actually elevate the fitness of normal intestinal stem cells to outcompete neighboring transformed APC mutant cancer cells ⁵. Several additional studies have now corroborated this finding ⁶. Nevertheless, we concur with the reviewer that a more detailed assessment of potential Wnt agonism of our molecules would strengthen our manuscript, particularly given that our global proteomics analysis of cancer cells treated with IGF1R PROTAB revealed an increase in Wnt pathway receptor components. This instigated us to further examine the effect of PROTABs on Wnt signaling potentiation. As such, we now present in our revised manuscript a comprehensive analysis of the impact of PROTAB on Wnt signaling both *in vitro* and *in vivo*. More specifically we have performed:

- 1. A characterization of our PROTAB in HEK293T cells that stably express a Wnt luciferase reporter (TOP-Flash). We find that indeed, our IGF1R PROTABs does lead to an increase in Wnt signaling activity, albeit of minimal impact compared to both RSPO3 treatment or GSK3beta inhibition lead to elevated Wnt activity (Extended data figure 3i).**
- 2. In line with the above, *in vivo* treatment with IGF1R PROTABs followed by transcriptional analysis of the whole gut led to only a minor increase in expression of genes related to Wnt signaling, stem cells and proliferation (Figure 3h).**
- 3. Importantly, long term treatment with PROTABs, at a concentration that leads to profound IGF1R degradation in tumors, had no discernible impact on intestinal architecture, demonstrating that the observed Wnt agonism does not translate into a hyperplastic phenotype in the gut (Figure 3i).**

Other remarks

-It is difficult to follow the experiment's jump between cell lines and models. For instance, it is clear that the methods are developed in the HEK293T cell line (easier to manipulate), and consequently validated in CRC cell lines like DLD1, HT115 and SW1417 among others. However,

it is rather confusing to present the results of the validation in a different line depending on the figure and the experiment (for example Fig. 2f, 2h, 3c and 3e all represent different lines).

This point is well taken and where appropriate we have clarified the rationale for the usage of distinct cell lines in various contexts in order to make the transition between cell lines easier to follow.

- 1. We have used more than 12 cell lines in order to validate the PROTAB technology, and in the revised version we extended models to organoids in order to orthogonally validate the technology across a large compendium of models**
- 2. HT29 has been extensively validated and optimized for CRISPR Cas9, therefore it was used for gene editing to ectopically express RNF43 and ZNRF3, as well as generate the HT29-HiBiT cells.**
- 3. We have used SW48 for *in vitro* assays, including pathway analysis and cell viability after PROTAB treatment, since we obtained substantial IGF1R degradation in this model. In addition, this cell line expresses PD-L1 and HER2, which enable expansion and validation of PROTAB activity to various receptors. Furthermore, SW48 grafted efficiently in mice, which was ideal for the *in vivo* evaluation of PROTAB activity.**
- 4. Long term inhibition of the proteasome pathway is highly toxic in numerous cell lines, therefore we used DLD1 and SW48 that tolerated treatment for up to 14 hours with proteasome inhibitors with minimal impact on cell death. The revised manuscript contains the small molecule inhibition data for SW48 (Figure 3 d, f and Extended data figure 4 d, e)**
- 5. In the revised version, we have added the ASPC1 pancreatic cancer cell line (Figure 4e), due to detectable Fzd expression compared to other CRC lines used in the paper**

-Why did the author select IGF1R as target of their repurposed transmembrane E3 ubiquitin ligases? In the context of CRC, the impact of IGF1R on CRC biology is rather unclear. Other transmembrane proteins (such as EGFR, HER2, PD-L1 and RNF43 C-term truncated mutants) are much more obvious candidates. IGF1R is more commonly implied in tumors of the liver (which have high frequencies of CTNNB1 mutations). Perhaps that would be a good model system to evaluate the clinical potential of the PROTABs targeting IGF1R.

As a matter of fact, we purposely selected IGF1R as a proof of concept because few efforts had been made to degrade this protein and various degradation approaches have been established for EGFR and PD-L1. We agree with this reviewer that IGF1R has not been identified as a driver of CRC, hence we merely used it as a proof of concept to demonstrate the potential of the PROTAB platform. Nevertheless, we do highlight some of the impact of degradation using PROTAB over classical antibody blocking, including deeper pathway

inhibition (Figure 2f) which is accompanied by increased efficacy of PROTAB in particular at low dose of antibody (Figure 2g). Per this reviewer's comment, we have now demonstrated degradation of additional receptors, including HER2, and PD-L1 in primary CRC organoids and also *in vivo* (Figure 4 a, b and Extended data figure 4 b).

-Why did the authors select CRC-derived immortalized cell lines as model system to evaluate the efficacy of their technique *in vitro*? Despite being easy to manipulate and culture, their reliability as relevant pre-clinical model remains questionable. In Fig.1 the authors present a pilot experiment performed on mouse organoids engineered to express multiple mutations found during CRC progression. Having access to such model systems can be fundamental to evaluate the efficacy of the technique in more relevant settings (CRC patients-derived organoids).

We thank this reviewer for raising this point. We certainly concur that validation of our PROTAB strategy in preclinical models that more faithfully recapitulate human disease will greatly improve on the impact of our manuscript. In our revised version we have now tested a series of human primary derived organoids (Figure 2j and Extended data figure 4b). In line with what was observed in various cell lines, we detected profound degradation in a variety of cancer derived organoids while minimal degradation was observed in normal colon organoids. Hence, we believe the organoid data further exemplify the tumor targeted strategy that is associated with the ZNRF3 PROTAB.

-Why was the antibody Cixutumumab selected for the targeting of the ECD of IGF1R and not Roba? According to the Western Blot in Fig. 1h and 1i, Roba displays a better clearance of IGF1R from the membrane.

We thank the reviewer for raising this point. We focused on Cixutumumab since its pairing with gD lead to appreciable yet incomplete IGF1R degradation. As such, it offered the possibility to identify ligase antibodies that further enhance degradation in an effort to further establish the optimal rules (i.e affinity of the ligase arm, geometry and valency) that govern degradation. In addition, we found that Roba displayed deep degradation as a bivalent and hence would have complicated the interpretation of the PROTAB further (Rebuttal figure 3).

Rebuttal figure 3. Western blot analysis of lysates from SW48 that were treated with the indicated bivalent antibodies. Endogenous IGF1R- β protein levels were detected. α -TUBULIN was used as a loading control. Data are representative of three independent experiments.

-Occasional degradation of the IGF1R is observed for the bivalent Cixutumumab formation, which is enhanced in the DLD1, KM12 and the HT115 lines. Is this effect comparable if using any of the other IGF1R targeting antibodies?

Per this reviewer's previous comment, we have evaluated all IGF1R bivalent antibodies and report on their impact on IGF1R degradation in Rebuttal figure 3.

-Figure 2i, was the injection of SW48 subcutaneous or orthotopic? Was any adverse effect observed during treatment with the PROTABs? Did the authors evaluate the inhibition of downstream effectors (like pAKT) as well after the in vivo delivery of the PROTABs?

We have performed subcutaneous injections. No adverse effects were observed. In particular we have treated animals for over a week with no evidence of weight loss or histological defects in the gut. This is now reported in Extended data figure 2k and Figure 3i.

-Blocking either the proteasomal or lysosomal protein degradation systems results in partial rescue of the PROTABs-induced IGF1R degradation (Fig. 3g). To the authors knowledge, does IGF1R possess a recycling mechanism to the cell surface for the non-degraded protein? Such mechanism is known for other membrane receptors like EGFR which recycles back to the membrane after escaping lysosomal-mediated degradation.

IGF1R endosomal trafficking has not been as extensively documented as the case of EGFR, yet there are several reports that support IGF1R recycling to the cell surface after bypassing lysosomal mediated degradation ⁷. In addition, both the proteasome and lysosome are required for IGF1R degradation ^{8,9}, which we also found to be the case after IGF1R PROTAB treatment. Furthermore, we now report that PD-L1 PROTAB also resulted in profound PD-L1 degradation, which was dependent on the proteasomal and lysosomal pathways (Extended figure 4e). Although this dual degradation pathways could be a general

feature of the PROTAB technology, further studies will be needed to fully understand the molecular mechanism that govern internalization, recycling, and route of degradation of receptors after PROTAB treatment.

-Fig. 4c, why the amount of IGF1R detected in the negative controls is significantly different in different PROTABs (for example RNF43 vs RNF149)? Also, why a double band is detected in RNF150 of the same figure?

We thank the reviewer for this comment. It is important to note that it was not technically feasible to run all the samples for the different ligases on the same gel. As such, IGF1R protein levels should only be compared between treatment groups for each ligase-specific cell line generated. This is why we present the ligase cell lines as separate boxed western blots to direct the reader to which samples can be objectively compared to one another. Regarding the ligases, we would like to firstly highlight that the ligases are of varying molecular weights, which can be appreciated in the full blots presented in Extended data figure 6 and 7. Regarding the double band observed with RNF149, one explanation could be that the ectopically expressed protein is processed or undergoes post translational modification that can be detected using the FLAG antibody. This is more apparent in the newly generated SW48 gD-ligases-FLAG cell lines used to asses PD-L1 degradation (Extended data figure 5e).

-Fig.4e, the degradation of PD-L1 is not clear from the figure. Can the authors present a more convincing picture?

We apologize for the poor quality of the original figure, which was mainly due to the low expression of PD-L1 in this particular cell line. We have further engineered all newly identified cell surface ligases in the SW48 cell line and evaluated PD-L1 degradation in this model. We have replaced the original figure 4c with the new set of data in Extended data figure 5e). Additionally, we have now demonstrated endogenous degradation of PD-L1 in SW48 using ZNRF3-based PD-L1 PROTABs (Extended data figure 4c).

-In general, all figures including Western Blots data should be presented with the original pictures as source data in the supplementary material attached to the main manuscript (For example, in Fig. 4e there is a band that has been cut from the picture).

To address the reviewer's comment, we now include the uncropped blots in Extended data figure 6 and 7.

References:

- 1 Korolchuk, V. I., Menzies, F. M. & Rubinsztein, D. C. Mechanisms of cross-talk between the ubiquitin-proteasome and autophagy-lysosome systems. *FEBS Lett* **584**, 1393-1398, doi:10.1016/j.febslet.2009.12.047 (2010).
- 2 van Kerkhof, P. *et al.* Proteasome inhibitors block a late step in lysosomal transport of selected membrane but not soluble proteins. *Mol Biol Cell* **12**, 2556-2566, doi:10.1091/mbc.12.8.2556 (2001).
- 3 Burtrum, D. *et al.* A fully human monoclonal antibody to the insulin-like growth factor I receptor blocks ligand-dependent signaling and inhibits human tumor growth in vivo. *Cancer Res* **63**, 8912-8921 (2003).
- 4 Koo, B. K. *et al.* Tumour suppressor RNF43 is a stem-cell E3 ligase that induces endocytosis of Wnt receptors. *Nature* **488**, 665-669, doi:10.1038/nature11308 (2012).
- 5 van Neerven, S. M. *et al.* Apc-mutant cells act as supercompetitors in intestinal tumour initiation. *Nature* **594**, 436-441, doi:10.1038/s41586-021-03558-4 (2021).
- 6 Flanagan, D. J. *et al.* NOTUM from Apc-mutant cells biases clonal competition to initiate cancer. *Nature* **594**, 430-435, doi:10.1038/s41586-021-03525-z (2021).
- 7 Segretin, M. E., Galeano, A., Roldan, A. & Schillaci, R. Insulin-like growth factor-1 receptor regulation in activated human T lymphocytes. *Horm Res* **59**, 276-280, doi:10.1159/000070625 (2003).
- 8 Carelli, S., Di Giulio, A. M., Paratore, S., Bosari, S. & Gorio, A. Degradation of insulin-like growth factor-I receptor occurs via ubiquitin-proteasome pathway in human lung cancer cells. *J Cell Physiol* **208**, 354-362, doi:10.1002/jcp.20670 (2006).
- 9 Girnita, L., Girnita, A. & Larsson, O. Mdm2-dependent ubiquitination and degradation of the insulin-like growth factor 1 receptor. *Proc Natl Acad Sci U S A* **100**, 8247-8252, doi:10.1073/pnas.1431613100 (2003).

Reviewer Reports on the First Revision:

Referees' comments:

Referee #2 (Remarks to the Author):

The authors have done an excellent job responding to the criticisms, concerns and comments of the reviewers and this manuscript is substantially improved. It is particularly exciting to see the demonstration of tumor specific degradation in vivo and the enhanced demonstration of substrate/ligase scope.

2 minor points:

Notes from editing process left in legend for Fig.1

There are 2 flag bands in the Ex Fig 3E and it appears that ZNF3-6 works independently of the Ring domain, bringing the mechanism of action into concern.

Referee #4 (Remarks to the Author):

In this manuscript, the authors describe the PROTAC technique to mediate the degradation of transmembrane receptors by repurposing E3 ubiquitin ligases towards non-conventional targets. The take advantage of two WNT-related E3 ligases known to be upregulated in CRC and show repurposing towards IGF1R, HER2 and PD-L1.

In this revised version of their manuscript the authors addressed most of the concerns raised at first submission. In particular, the authors demonstrate superior specificity of the PROTABs by comparing the degradation of the target between normal and tumor cells with a variety of approaches (including more relevant model systems). The updated results about this comparison (regarding IGF1R clearance between normal and tumor cells) are quite striking. Also, they resolved the concerns regarding the potential overactivation of WNT pathway in normal intestinal cells both in vitro and in vivo. Finally, the authors demonstrate the activity of PROTABs towards two additional targets (HER2 and PD-L1) both in vitro and in vivo.

Taken together, I think that the authors have put together considerable efforts to answer the most important remarks, with results that support their conclusion.

One remaining concern is the difference in activity observed when comparing different targeted proteins (for example the clearance of IGF1R, HER2 and PD-L1) which challenges the broad applicability of this technology in the context of tumor biology.

Author Rebuttals to First Revision:

Response to reviewers' comments for Marei *et al.* "Repurposing E3 ubiquitin ligases as cell surface protein degraders using Proteolysis Targeting Antibodies" Nature ID manuscript # 2021-11-17506A

REVIEWER #2

The authors have done an excellent job responding to the criticisms, concerns and comments of the reviewers and this manuscript is substantially improved. It is particularly exciting to see the demonstration of tumor specific degradation in vivo and the enhanced demonstration of substrate/ligase scope.

- **We thank the reviewer for helping us improve on our first submission and are pleased that the reviewer is satisfied with our revised manuscript.**

2 minor points:

Notes from editing process left in legend for Fig.1

- **We appreciate the thorough review. We have corrected the figure legend accordingly.**

There are 2 flag bands in the Ex Fig 3E and it appears that ZNF3-6 works independently of the Ring domain, bringing the mechanism of action into concern.

- **We thank the reviewer for their comment. Regarding the 2 FLAG bands, we occasionally observe a number of non-specific bands when probing ectopic protein expression using the FLAG M2 antibody, particularly with prolonged exposure. As highlighted below in the uncropped blot referring to Extended data figure 3 e, the high molecular weight band is also detected in the parental cell line that lacks exogenous ligase expression (uncropped blot is depicted in Supplementary Info File 2). In agreement with this being a non-specific band rather than a biologically meaningful ligase modification, the band was not observed in Extended data figure 3 f (see corresponding uncropped blot in Supplementary Info File 2), in which the lysates were more concentrated and thus extended exposure of the blots was not needed.**

Rebuttal figure 1. Uncropped Western blot scans for panels in Extended data figure 3 e

- **Regarding the reviewer's comment on the mechanism of action, we would like to note that the key conclusion that can be made from data presented in Extended data figure 3 e is that overexpression of the wild type ligase drives PROTAB-mediated degradation to a greater extent than the delta RING mutant. The tested PROTABs can also engage endogenous ZNRF3 and this could potentially explain the variable baseline observed in the parental line. This does not undermine the central finding, but it does create complications in comparing activities across the different PROTABs. Nevertheless, we do observe enhanced IGF1R clearance in ZNRF3 WT expressing cells treated with PROTABs compared to the ZNRF3 delta RING expressing cells, even with the ZNRF3-6 based PROTAB, although to a lesser extent. This result is also supported with the inability of the endogenous ZNRF3 RING(i) (indel) to degrade IGF1R upon treatment with the ZNRF3-6*IGF1R PROTAB outlined in Figure 3 c.**
- **We have included additional clarification in the results section.**

REVIEWER #4

In this manuscript, the authors describe the PROTAC technique to mediate the degradation of transmembrane receptors by repurposing E3 ubiquitin ligases towards non-conventional targets. The take advantage of two WNT-related E3 ligases known to be upregulated in CRC and show repurposing towards IGF1R, HER2 and PD-L1.

In this revised version of their manuscript the authors addressed most of the concerns raised at first submission. In particular, the authors demonstrate superior specificity of the PROTABs by comparing the degradation of the target between normal and tumor cells with a variety of approaches (including more relevant model systems). The updated results about this comparison (regarding IGF1R clearance between normal and tumor cells) are quite striking. Also, they resolved the concerns regarding the potential overactivation of WNT pathway in normal intestinal cells both in vitro and in vivo. Finally, the authors demonstrate the activity of PROTABs towards two additional targets (HER2 and PD-L1) both in vitro and in vivo.

Taken together, I think that the authors have put together considerable efforts to answer the most important remarks, with results that support their conclusion.

- **We thank the reviewer for their enthusiasm regarding our revised manuscript and its potential impact on the scientific community.**

One remaining concern is the difference in activity observed when comparing different targeted proteins (for example the clearance of IGF1R, HER2 and PD-L1) which challenges the broad applicability of this technology in the context of tumor biology.

- **We thank the reviewer for their thoughtful perspective on the manuscript. We agree that as an off-the-shelf technology PROTABs are unlikely to drive profound degradation of every target. The requirement for an intracellular domain, reduced activity on receptors with elevated copy number, and a reliance on the inherent ubiquitination machinery and its associated regulation may limit the general utility of this approach. That said, we do believe PROTABs have the potential for broad, but not universal, applicability. The current manuscript provides evidence supporting the usage of PROTABs across various ligases and substrates, both *in vitro* and *in vivo* (Figure 4 a, b, e; Extended data figure 4; Extended data figure 5 d, e). We believe the observed variability highlights the need for further optimization of PROTABs on a target/ligase basis, similar to the history of ligand and linker optimization for the PROTACs. To this end, we have defined multiple approaches to enhance the degradative potential of the antibodies including selecting component binders with high affinity and appropriate epitope (Figure 2 a-c) and engineering different PROTAB formats (Figure 4 f-j). We also note the diverse expression patterns of multiple cell surface ligases (Extended data figure 5 f), which could provide the opportunity for “ligase-shopping” to pair selective ligase expression to the disease of interest. We believe that the opportunity for selective degradation and the wide scope of available ligases makes this technology compare favorably to related technologies and that there will be multiple opportunities to apply the technology both in tumor biology as well as other pathological conditions.**
- **We have included this perspective in the summary section.**